# Weakly Supervised Clustering by Exploiting Unique Class Count

**Mustafa Umit Oner**[1,2]**, Hwee Kuan Lee**[1,2,3,4] **& Wing-Kin Sung**[1,5]
[1]School of Computing, National University of Singapore, Singapore 117417, [2]A*STAR
Bioinformatics Institute, Singapore 138671, [3]Image and Pervasive Access Lab (IPAL),
CNRS UMI 2955, Singapore 138632, [4]Singapore Eye Research Institute, Singapore
169856, [5]A*STAR Genome Institute of Singapore, Singapore 138672
`{umitoner,ksung}@comp.nus.edu.sg, {leehk}@bii.a-star.edu.sg`

## Abstract

A weakly supervised learning based clustering framework is proposed in this paper. As the core of this framework, we introduce a novel multiple instance learning task based on a bag level label called unique class count ($ucc$), which is the number of unique classes among all instances inside the bag. In this task, no annotations on individual instances inside the bag are needed during training of the models. We mathematically prove that with a perfect $ucc$ classifier, perfect clustering of individual instances inside the bags is possible even when no annotations on individual instances are given during training. We have constructed a neural network based $ucc$ classifier and experimentally shown that the clustering performance of our framework with our weakly supervised $ucc$ classifier is comparable to that of fully supervised learning models where labels for all instances are known. Furthermore, we have tested the applicability of our framework to a real world task of semantic segmentation of breast cancer metastases in histological lymph node sections and shown that the performance of our weakly supervised framework is comparable to the performance of a fully supervised Unet model.

## 1 Introduction

In machine learning, there are two main learning tasks on two ends of scale bar: unsupervised learning and supervised learning. Generally, performance of supervised models is better than that of unsupervised models since the mapping between data and associated labels is provided explicitly in supervised learning. This performance advantage of supervised learning requires a lot of labelled data, which is expensive. Any other learning tasks reside in between these two tasks, so are their performances. Weakly supervised learning is an example of such tasks. There are three types of supervision in weakly supervised learning: *incomplete*, *inexact* and *inaccurate* supervision. Multiple instance learning (MIL) is a special type of weakly supervised learning and a typical example of inexact supervision (Zhou, 2017). In MIL, data consists of bags of instances and their corresponding bag level labels. Although the labels are somehow related to instances inside the bags, the instances are not explicitly labeled. In traditional MIL, given the bags and corresponding bag level labels, task is to learn the mapping between bags and labels while the goal is to predict labels of unseen bags (Dietterich et al., 1997; Foulds & Frank, 2010).

In this paper, we explore the feasibility of finding out labels of individual instances inside the bags only given the bag level labels, i.e. there is no individual instance level labels. One important application of this task is semantic segmentation of breast cancer metastases in histological lymph node sections, which is a crucial step in staging of breast cancer (Brierley et al., 2016). In this task, each pathology image of a lymph node section is a bag and each pixel inside that image is an instance. Then, given the bag level label that whether the image contains metastases or not, the task is to label each pixel as either metastases or normal. This task can be achieved by asking experts to exhaustively annotate each metastases region in each image. However, this exhaustive annotation process is tedious, time consuming and more importantly not a part of clinical workflow.

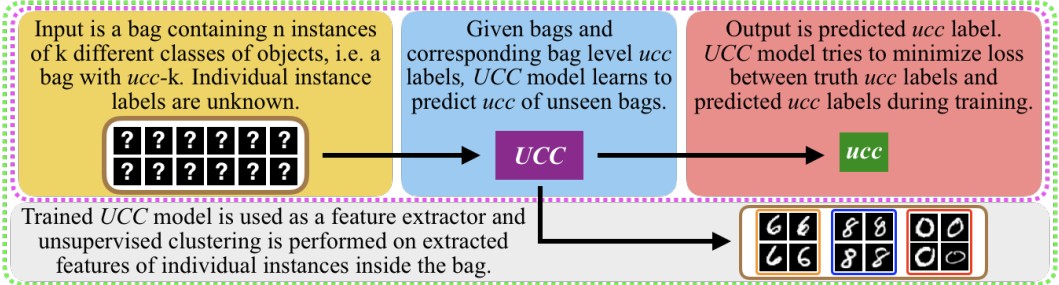

Figure 1: Weakly supervised clustering framework. Our framework (green dashed line) consists of the $UCC$ model (magenta dashed line) and the unsupervised instance clustering branch.

In many complex systems, such as in many types of cancers, measurements can only be obtained at coarse level (bag level), but information at fine level (individual instance level) is of paramount importance. To achieve this, we propose a weakly supervised learning based clustering framework. Given a dataset consisting of instances with unknown labels, our ultimate objective is to cluster the instances in this dataset. To achieve this objective, we introduce a novel MIL task based on a new kind of bag level label called unique class count ($ucc$), which is the number of unique classes or the number of clusters among all the instances inside the bag. We organize the dataset into non-empty bags, where each bag is a subset of individual instances from this dataset. Each bag is associated with a bag level $ucc$ label. Then, our MIL task is to learn mapping between the bags and their associated bag level $ucc$ labels and then to predict the $ucc$ labels of unseen bags. We mathematically show that a $ucc$ classifier trained on this task can be used to perform unsupervised clustering on individual instances in the dataset. Intuitively, for a $ucc$ classifier to count the number of unique classes in a bag, it has to first learn discriminant features for underlying classes. Then, it can group the features obtained from the bag and count the number of groups, so the number of unique classes.

Our weakly supervised clustering framework is illustrated in Figure 1. It consists of a neural network based $ucc$ classifier, which is called as *Unique Class Count (UCC)* model, and an unsupervised clustering branch. The $UCC$ model accepts any bag of instances as input and uses $ucc$ labels for supervised training. Then, the trained $UCC$ model is used as a feature extractor and unsupervised clustering is performed on the extracted features of individual instances inside the bags in the clustering branch. One application of our framework is the semantic segmentation of breast cancer metastases in lymph node sections (see Figure 4). The problem can be formulated as follows. The input is a set of images. Each image (bag) has a label of $ucc1$ (image is fully normal or fully metastases) or $ucc2$ (image is a mixture of normal and metastases). Our aim is to segment the pixels (instances) in the image into normal and metastases. A $UCC$ model can be trained to predict $ucc$ labels of individual images in a fully supervised manner; and the trained model can be used to extract features of pixels (intances) inside the images (bags). Then, semantic segmentation masks can be obtained by unsupervised clustering of the pixels (each is represented by the extracted features) into two clusters (metastases or normal). Note that $ucc$ does not directly provide an exact label for each individual instance. Therefore, our framework is a weakly supervised clustering framework.

Finally, we have constructed $ucc$ classifiers and experimentally shown that clustering performance of our framework with our $ucc$ classifiers is better than the performance of unsupervised models and comparable to performance of fully supervised learning models. We have also tested the performance of our model on the real world task of semantic segmentation of breast cancer metastases in lymph node sections. We have compared the performance of our model with the performance of popular medical image segmentation architecture of $Unet$ (Ronneberger et al., 2015) and shown that our weakly supervised model approximates the performance of fully supervised $Unet$ model[1].

Hence, there are three main contributions of this paper:

1. We have defined *unique class count* as a bag level label in MIL setup and mathematically proved that a perfect $ucc$ classifier, in principle, can be used to perfectly cluster the individual instances inside the bags.

---

[1] **Code and trained models:** http://bit.ly/uniqueclasscount

2. We have constructed a neural network based $ucc$ classifier by incorporating kernel density estimation (KDE) (Parzen, 1962) as a layer into our model architecture, which provided us with end-to-end training capability.

3. We have experimentally shown that clustering performance of our framework is better than the performance of unsupervised models and comparable to performance of fully supervised learning models.

The rest of the paper is organized such that related work is in Section 2, details of our weakly supervised clustering framework are in Section 3, results of the experiments on MNIST, CIFAR10 and CIFAR100 datasets are in Section 4, results of the experiments in semantic segmentation of breast cancer metastases are in Section 5, and Section 6 concludes the paper.

## 2 RELATED WORK

This work is partly related to MIL which was first introduced in (Dietterich et al., 1997) for drug activity prediction. Different types of MIL were derived with different assumptions (Gärtner et al., 2002; Zhang & Goldman, 2002; Chen et al., 2006; Foulds, 2008; Zhang & Zhou, 2009; Zhou et al., 2009), which are reviewed in detail in (Foulds & Frank, 2010), and they were used for many different applications such as, image annotation/categorization/retrieval (Chen & Wang, 2004; Zhang et al., 2002; Tang et al., 2010), text categorization (Andrews et al., 2003; Settles et al., 2008), spam detection (Jorgensen et al., 2008), medical diagnosis (Dundar et al., 2007), face/object detection (Zhang et al., 2006; Felzenszwalb et al., 2010) and object tracking (Babenko et al., 2011).

In MIL, different types of pooling layers are used to combine extracted features of instances inside the bags, such as max-pooling and log-sum-exp pooling (Ramon & De Raedt, 2000; Zhou & Zhang, 2002; Wu et al., 2015; Wang et al., 2018). On the other hand, our $UCC$ model uses KDE layer in order to estimate the distribution of extracted features. The advantage of KDE over pooling layers is that it embeds the instance level features into distribution space rather than summarizing them.

There are also methods modeling cardinality and set distributions (Liu et al., 2015; Brukhim & Globerson, 2018; Kipf et al., 2018). However, cardinality of a set and $ucc$ are completely different from each other. It is also important to state that $ucc$ is obviously different from object/crowd counting (Idrees et al., 2013; Arteta et al., 2014; Zhang et al., 2015; 2016) since the task in object/crowd counting is to count the instances of the same type of object or people.

Lastly, we compare clustering accuracies of our models with clustering accuracies of unsupervised baseline models: K-means (Wang et al., 2015) and Spectral Clustering (Zelnik-Manor & Perona, 2005); state of the art unsupervised models: JULE (Yang et al., 2016), GMVAE (Dilokthanakul et al., 2016), DAC (Chang et al., 2017), DEPICT (Ghasedi Dizaji et al., 2017) and DEC (Xie et al., 2016); and state of the art semi-supervised models: AAE (Makhzani et al., 2015), CatGAN (Springenberg, 2015), LN (Rasmus et al., 2015) and ADGM (Maaløe et al., 2016).

## 3 WEAKLY SUPERVISED CLUSTERING FRAMEWORK

In this section, we state our machine learning objective and formally define our novel MIL task, which is the core of our weakly supervised clustering framework. Finally, we explain details of the two main components of our framework, namely $UCC$ model and unsupervised clustering branch.

**Objective:** Let $\mathcal{X} = \{x_1, x_2, \cdots, x_n\}$ be a dataset such that each instance $x_i \in \mathcal{X}$ belongs to a class, but its label is unknown. In this paper, we assume that total number of classes $K$ is known. Hence, each instance $x_i$ is endowed with an underlying, but unkown, label $\mathcal{L}(x_i) = l_i \in \{1, 2, \cdots, K\}$. Further assume that for each class $k \in \{1, 2, \cdots K\}$, there exist at least one element $x_i \in \mathcal{X}$ such that $\mathcal{L}(x_i) = l_i = k$. Our eventual objective is to derive a predicted class label $\hat{l}_i$ for each instance $x_i$ that tends towards underlying truth class $l_i$, i.e. $\hat{l}_i \to \mathcal{L}(x_i) = l_i$.

### 3.1 A NOVEL MIL TASK

In this novel MIL task, *unique class count* is used as an inexact, weak, bag level label and is defined in Definition 1. Assume that we are given subsets $\sigma_\zeta \subset \mathcal{X}, \zeta = 1, 2, \cdots, N$ and unique class counts

$\eta_{\sigma_\zeta} \forall \sigma_\zeta$. Hence, MIL dataset is $\mathcal{D} = \{(\sigma_1, \eta_{\sigma_1}), \cdots, (\sigma_N, \eta_{\sigma_N})\}$. Then, our MIL task is to learn the mapping between the bags and their associated bag level $ucc$ labels while the goal is to predict the $ucc$ labels of unseen bags.

**Definition 1** *Given a subset $\sigma_\zeta \subset \mathcal{X}$, unique class count, $\eta_{\sigma_\zeta}$, is defined as the number of unique classes that all instances in the subset $\sigma_\zeta$ belong to, i.e. $\eta_{\sigma_\zeta} = |\{\mathcal{L}(x_i)|x_i \in \sigma_\zeta\}|$. Recall that each instance belongs to an underlying unknown class.*

Given a dataset $\mathcal{D}$, our eventual objective is to assign a label to each instance $x_i \in \mathcal{X}$ such that assigned labels and underlying unknown classes are consistent. To achieve this eventual objective, a deep learning model is designed such that the following intermediate objectives can be achieved while it is being trained on our MIL task:

1. **Unique class count**: Given an unseen set $\sigma_\zeta$, the deep learning model, which is trained on $\mathcal{D}$, can predict its unique class count $\eta_{\sigma_\zeta}$ correctly.

2. **Labels on sets**: Let $\sigma_\zeta^{pure}$ and $\sigma_\xi^{pure}$ be two disjoint pure sets (Definition 2) such that while all instances in $\sigma_\zeta^{pure}$ belong to one underlying class, all instances in $\sigma_\xi^{pure}$ belong to another class. Given $\sigma_\zeta^{pure}$ and $\sigma_\xi^{pure}$, the deep learning model should enable us to develop an unsupervised learning model to label instances in $\sigma_\zeta^{pure}$ and $\sigma_\xi^{pure}$ as belonging to different classes. Note that the underlying classes for instances in the sets are unknown.

3. **Labels on instances**: Given individual instances $x_i \in \mathcal{X}$, the deep learning model should enable us to assign a label to each individual instance $x_i$ such that all instances with different/same underlying unknown classes are assigned different/same labels. This is the eventual unsupervised learning objective.

**Definition 2** *A set $\sigma$ is called a pure set if its unique class count equals one. All pure sets is denoted by the symbol $\sigma^{pure}$ in this paper.*

## 3.2 Unique Class Count Model

In order to achieve the stated objectives, we have designed a deep learning based *Unique Class Count (UCC)* model. Our $UCC$ model consists of three neural network modules ($\theta_{\text{feature}}, \theta_{\text{drn}}, \theta_{\text{decoder}}$) and can be trained end-to-end. The first module $\theta_{\text{feature}}$ extracts features from individual instances; then distributions of features are constructed from extracted features. The second module $\theta_{\text{drn}}$ is used to predict $ucc$ label from these distributions. The last module $\theta_{\text{decoder}}$ is used to construct an autoencoder together with $\theta_{\text{feature}}$ so as to improve the extracted features by ensuring that extracted features contain semantic information for reconstruction.

Formally, for $x_i \in \sigma_\zeta, i = \{1, 2, \cdots, |\sigma_\zeta|\}$, feature extractor module $\theta_{\text{feature}}$ extracts $J$ features $\{f_{\sigma_\zeta}^{1,i}, f_{\sigma_\zeta}^{2,i}, \cdots, f_{\sigma_\zeta}^{J,i}\} = \theta_{\text{feature}}(x_i)$ for each instance $x_i \in \sigma_\zeta$. As a short hand, we write the operator $\theta_{\text{feature}}$ as operating element wise on the set to generate a feature matrix $\theta_{\text{feature}}(\sigma_\zeta) = f_{\sigma_\zeta}$ with matrix elements $f_{\sigma_\zeta}^{j,i} \in \mathbb{R}$, representing the $j^{th}$ feature of the $i^{th}$ instance. After obtaining features for all instances in $\sigma_\zeta$, a kernel density estimation (KDE) module is used to accumulate feature distributions $h_{\sigma_\zeta} = (h_{\sigma_\zeta}^1(v), h_{\sigma_\zeta}^2(v), \cdots, h_{\sigma_\zeta}^J(v))$. Then, $h_{\sigma_\zeta}$ is used as input to distribution regression module $\theta_{\text{drn}}$ to predict the $ucc$ label, $\tilde{\eta}_{\sigma_\zeta} = \theta_{\text{drn}}(h_{\sigma_\zeta})$ as a softmax vector $(\tilde{\eta}_{\sigma_\zeta}^1, \tilde{\eta}_{\sigma_\zeta}^2, \cdots, \tilde{\eta}_{\sigma_\zeta}^K)$. Concurrently, decoder module $\theta_{\text{decoder}}$ in autoencoder branch is used to reconstruct the input images from the extracted features in an unsupervised fashion, $\tilde{x}_i = \theta_{\text{decoder}}(\theta_{\text{feature}}(x_i))$. Hence, $UCC$ model, main modules of which are illustrated in Figure 2(a), optimizes two losses concurrently: 'ucc loss' and 'autoencoder loss'. While 'ucc loss' is cross-entropy loss, 'autoencoder loss' is mean square error loss. Loss for one bag is given in Equation 1.

$$\alpha \underbrace{\left[\sum_{k=1}^{K} \eta_{\sigma_\zeta}^k \log \tilde{\eta}_{\sigma_\zeta}^k\right]}_{ucc\ loss} + (1-\alpha) \underbrace{\left[\frac{1}{|\sigma_\zeta|} \sum_{i=1}^{|\sigma_\zeta|} (x_i - \tilde{x}_i)^2\right]}_{autoencoder\ loss} \quad \text{where } \alpha \in [0,1] \tag{1}$$

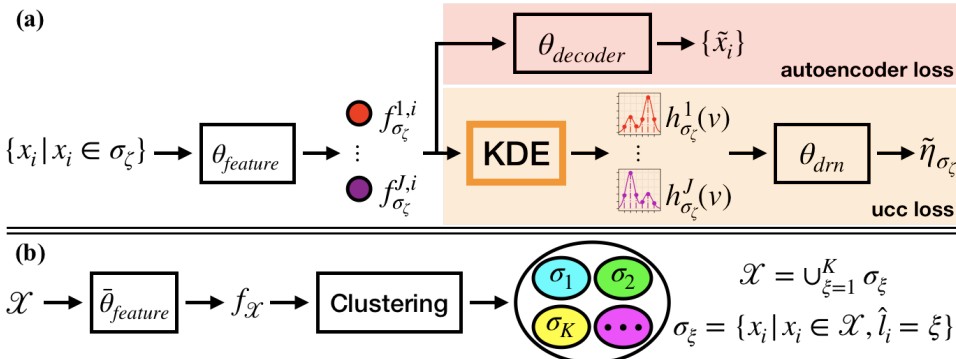

Figure 2: Weakly supervised clustering framework. **(a) *UCC* model:** $\theta_{\text{feature}}$ extracts $J$ features, shown in colored nodes. KDE module obtains feature distribution for each feature. Then, $\theta_{\text{drn}}$ predicts the *ucc* label $\tilde{\eta}_{\sigma_\zeta}$. Concurrently, decoder module $\theta_{\text{decoder}}$ in autoencoder branch reconstructs the input images from the extracted features. **(b) Unsupervised clustering:** Trained feature extractor, $\bar{\theta}_{\text{feature}}$, is used to extract the features of all instances in $\mathcal{X}$ and unsupervised clustering is performed on extracted features. Note that $\hat{l}_i$ is clustering label of $x_i \in \mathcal{X}$.

### 3.2.1 KERNEL DENSITY ESTIMATION MODULE

In $UCC$ model, input is a set $\sigma_\zeta$ and output is corresponding *ucc* label $\tilde{\eta}_{\sigma_\zeta}$, which does not depend on permutation of the instances in $\sigma_\zeta$. KDE module provides $UCC$ model with permutation-invariant property. Moreover, KDE module uses the Gaussian kernel and it is differentiable, so our model can be trained end-to-end (Appendix A). KDE module also enables our theoretical analysis thanks to its decomposability property (Appendix B). Lastly, KDE module estimates the probability distribution of extracted features and enables $\theta_{\text{drn}}$ to fully utilize the information in the shape of the distribution rather than looking at point estimates of distribution obtained by other types of pooling layers (Ramon & De Raedt, 2000; Zhou & Zhang, 2002; Wang et al., 2018) (Appendix C.6).

### 3.2.2 PROPERTIES OF UNIQUE CLASS COUNT MODEL

This section mathematically proves that the $UCC$ model guarantees, in principle, to achieve the stated intermediate objectives in Section 3.1. Proof of propositions are given in Appendix B.

**Proposition 1** *Let $\sigma_\zeta$, $\sigma_\xi$ be disjoint subsets of $\mathcal{X}$ with predicted unique class counts $\tilde{\eta}_{\sigma_\zeta} = \tilde{\eta}_{\sigma_\xi} = 1$. If the predicted unique class count of $\sigma_\nu = \sigma_\zeta \cup \sigma_\xi$ is $\tilde{\eta}_{\sigma_\nu} = 2$, then $h_{\sigma_\zeta} \neq h_{\sigma_\xi}$.*

**Definition 3** *A perfect unique class count classifier takes in any set $\sigma$ and output the correct predicted unique class count $\tilde{\eta}_\sigma = \eta_\sigma$.*

**Proposition 2** *Given a perfect unique class count classifier. The dataset $\mathcal{X}$ can be perfectly clustered into $K$ subsets $\sigma_\xi^{pure}, \xi = 1, 2, \cdots, K$, such that $\mathcal{X} = \bigcup_{\xi=1}^{K} \sigma_\xi^{pure}$ and $\sigma_\xi^{pure} = \{x_i | x_i \in \mathcal{X}, \mathcal{L}(x_i) = \xi\}$.*

**Proposition 3** *Given a perfect unique class count classifier. Decompose the dataset $\mathcal{X}$ into $K$ subsets $\sigma_\xi^{pure}, \xi = 1, \cdots K$, such that $\sigma_\xi^{pure} = \{x_i | x_i \in \mathcal{X}, \mathcal{L}(x_i) = \xi\}$. Then, $h_{\sigma_\xi^{pure}} \neq h_{\sigma_\zeta^{pure}}$ for $\xi \neq \zeta$.*

Suppose we have a perfect *ucc* classifier. For any two pure sets $\sigma_\zeta^{pure}$ and $\sigma_\xi^{pure}$, which consist of instances of two different underlying classes, *ucc* labels must be predicted correctly by the perfect *ucc* classifier. Hence, the conditions of Proposition 1 are satisfied, so we have $h_{\sigma_\zeta^{pure}} \neq h_{\sigma_\xi^{pure}}$. Therefore, we can, in principle, perform an unsupervised clustering on the distributions of the sets without knowing the underlying truth classes of the instances. Hence, the perfect *ucc* classifier enables us to achieve our intermediate objective of **"Labels on sets"**. Furthermore, given a perfect *ucc* classifier, Proposition 2 states that by performing predictions of *ucc* labels alone, without any

knowledge of underlying truth classes for instances, one can in principle perform perfect clustering for individual instances. Hence, a perfect $ucc$ classifier enables us to achieve our intermediate objective of **"Labels on instances"**.

### 3.3 Unsupervised Instance Clustering

In order to achieve our ultimate objective of developing an unsupervised learning model for clustering all the instances in dataset $\mathcal{X}$, we add this unsupervised clustering branch into our framework. Theoreticallly, we have shown in Proposition 3 that given a perfect $ucc$ classifier, distributions of pure subsets of instances coming from different underlying classes are different.

In practice, it may not be always possible (probably most of the times) to train a perfect $ucc$ classifier, so we try to approximate it. First of all, we train our $ucc$ classifier on our novel MIL task and save our trained model ($\bar{\theta}_{\text{feature}}, \bar{\theta}_{\text{drn}}, \bar{\theta}_{\text{decoder}}$). Then, we use trained feature extractor $\bar{\theta}_{\text{feature}}$ to obtain feature matrix $f_{\mathcal{X}} = \bar{\theta}_{\text{feature}}(\mathcal{X})$. Finally, extracted features are clustered in an unsupervised fashion, by using simple k-means and spectral clustering methods. Figure 2(b) illustrates the unsupervised clustering process in our framework. A good feature extractor $\bar{\theta}_{\text{feature}}$ is of paramount importance in this task. Relatively poor $\bar{\theta}_{\text{feature}}$ may result in a poor unsupervised clustering performance in practice even if we have a strong $\bar{\theta}_{\text{drn}}$. To obtain a strong $\bar{\theta}_{\text{feature}}$, we employ an autoencoder branch, so as to achieve high clustering performance in our unsupervised instance clustering task. The autoencoder branch ensures that features extracted by $\bar{\theta}_{\text{feature}}$ contain semantic information for reconstruction.

## 4 Experiments on MNIST and CIFAR Datasets

This section analyzes the performances of our $UCC$ models and fully supervised models in terms of our eventual objective of unsupervised instance clustering on MNIST (10 clusters) (LeCun et al., 1998), CIFAR10 (10 clusters) and CIFAR100 (20 clusters) datasets (Krizhevsky & Hinton, 2009).

### 4.1 Model Architectures and Datasets

To analyze different characteristics of our framework, different kinds of *unique class count* models were trained during our experiments: $UCC$, $UCC^{2+}$, $UCC_{\alpha=1}$ and $UCC^{2+}_{\alpha=1}$. These *unique class count* models took sets of instances as inputs and were trained on $ucc$ labels. While $UCC$ and $UCC^{2+}$ models had autoencoder branch in their architecture and they were optimized jointly over both *autoencoder loss* and *ucc loss*, $UCC_{\alpha=1}$ and $UCC^{2+}_{\alpha=1}$ models did not have autoencoder branch in their architecture and they were optimized over *ucc loss* only (i.e. $\alpha = 1$ in Equation 1). The aim of training *unique class count* models with and without autoencoder branch was to show the effect of autoencoder branch in the robustness of clustering performance with respect to $ucc$ classification performance. $UCC$ and $UCC_{\alpha=1}$ models were trained on bags with labels of $ucc1$ to $ucc4$. On the other hand, $UCC^{2+}$ and $UCC^{2+}_{\alpha=1}$ models were trained on bags with labels $ucc2$ to $ucc4$. Our models were trained on $ucc$ labels up to $ucc4$ instead of $ucc10$ ($ucc20$ in CIFAR100) since the performance was almost the same for both cases and training with $ucc1$ to $ucc4$ was much faster (Appendix C.2). Please note that for perfect clustering of instances inside the bags, it is enough to have a perfect $ucc$ classifier that can perfectly discriminate $ucc1$ and $ucc2$ bags from Proposition 2. The aim of traininig $UCC^{2+}$ and $UCC^{2+}_{\alpha=1}$ models was to experimentally check whether these models can perform as good as $UCC$ and $UCC_{\alpha=1}$ models even if there is no pure subsets during training. In addition to our *unique class count* models, for benchmarking purposes, we also trained fully supervised models, $FullySupervised$, and unsupervised autoencoder models, $Autoencoder$. $FullySupervised$ models took individual instances as inputs and used instance level ground truths as labels during training. On the other hand, $Autoencoder$ models were trained in an unsupervised manner by optimizing *autoencoder loss* (i.e. $\alpha = 0$ in Equation 1). It is important to note that all models for a dataset shared the same architecture for feature extractor module and all the modules in our models are fine tuned for optimum performance and training time as explained in Appendix C.1.

We trained and tested our models on MNIST, CIFAR10 and CIFAR100 datasets. We have $\mathcal{X}_{mnist,tr}$, $\mathcal{X}_{mnist,val}$ and $\mathcal{X}_{mnist,test}$ for MNIST; $\mathcal{X}_{cifar10,tr}$, $\mathcal{X}_{cifar10,val}$ and $\mathcal{X}_{cifar10,test}$ for CIFAR10; and $\mathcal{X}_{cifar100,tr}$, $\mathcal{X}_{cifar100,val}$ and $\mathcal{X}_{cifar100,test}$ for CIFAR100. Note that $tr$, $val$ and $test$ subscripts stand for 'training', 'validation' and 'test' sets, respectively. All the results presented in this paper were obtained on hold-out test sets $\mathcal{X}_{mnist,test}$, $\mathcal{X}_{cifar10,test}$ and $\mathcal{X}_{cifar100,test}$. $FullySupervised$

Table 1: Minimum inter-class JS divergence values, $ucc$ classification accuracy values and clustering accuracy values of our models (first part), baseline and state of the art unsupervised models (second part) and state of the art semi-supervised models (third part) on different test datasets. The best clustering accuracy values for each kind of models (weakly supervised (our models), unsupervised, semi-supervised) are highlighted in **bold**. ('x': not applicable, '-': missing')

| | min. JS divergence | | | $ucc$ acc. | | | clustering acc. | | |
|---|---|---|---|---|---|---|---|---|---|
| | mnist | cifar10 | cifar100 | mnist | cifar10 | cifar100 | mnist | cifar10 | cifar100 |
| $UCC$ | 0.222 | 0.097 | 0.004 | 1.000 | 0.972 | 0.824 | **0.984** | **0.781** | **0.338** |
| $UCC^{2+}$ | 0.251 | 0.005 | 0.002 | 1.000 | 0.936 | 0.814 | **0.984** | 0.545 | 0.278 |
| $UCC_{\alpha=1}$ | 0.221 | 0.127 | 0.003 | 1.000 | 0.982 | 0.855 | 0.981 | 0.774 | 0.317 |
| $UCC^{2+}_{\alpha=1}$ | 0.023 | 0.002 | 0.003 | 0.996 | 0.920 | 0.837 | 0.881 | 0.521 | 0.284 |
| $Autoencoder$ | 0.101 | 0.004 | 0.002 | x | x | x | 0.930 | 0.241 | 0.167 |
| $FullySupervised$ | 0.283 | 0.065 | 0.019 | x | x | x | 0.988 | 0.833 | 0.563 |
| JULE (Yang et al., 2016) | | | | | | | 0.964 | 0.272 | 0.137 |
| GMVAE (Dilokthanakul et al., 2016) | | | | | | | 0.885 | - | - |
| DAC (Chang et al., 2017)* | | | | | | | **0.978** | **0.522** | **0.238** |
| DEC (Xie et al., 2016)* | | | | | | | 0.843 | 0.301 | 0.185 |
| DEPICT (Ghasedi Dizaji et al., 2017)* | | | | | | | 0.965 | - | - |
| Spectral (Zelnik-Manor & Perona, 2005) | | | | | | | 0.696 | 0.247 | 0.136 |
| K-means (Wang et al., 2015) | | | | | | | 0.572 | 0.229 | 0.130 |
| ADGM (Maaløe et al., 2016) | | | | | | | **0.990** | - | - |
| Ladder Networks (Rasmus et al., 2015) | | | | | | | 0.989 | 0.796 | - |
| AAE (Makhzani et al., 2015) | | | | | | | 0.981 | - | - |
| CatGAN (Springenberg, 2015) | | | | | | | 0.981 | **0.804** | - |

\* Models do not separate training and testing data, i.e. their results are not on hold-out test sets.

models took individual instances as inputs and were trained on instance level ground truths. *Unique class count* models took sets of instances as inputs, which were sampled from the power sets $2^{\mathcal{X}_{mnist,tr}}$, $2^{\mathcal{X}_{cifar10,tr}}$ and $2^{\mathcal{X}_{cifar100,tr}}$, and were trained on $ucc$ labels (Appendix C.2). While all the models were trained in a supervised setup, either on $ucc$ labels or instance level ground truths, all of them were used to extract features for unsupervised clustering of individual instances.

## 4.2 Unique Class Count Prediction

Preceeding sections showed, in theory, that a perfect $ucc$ classifier can perform 'weakly' supervised clustering perfectly. We evaluate $ucc$ prediction accuracy of our *unique class count* models in accordance with our first intermediate objective that *unique class count* models should predict $ucc$ labels of unseen subsets correctly. We randomly sampled subsets for each $ucc$ label from the power sets of test sets and predicted the $ucc$ labels by using trained models. Then, we calculated the $ucc$ prediction accuracies by using predicted and truth $ucc$ labels, which are summarized in Table 1 (Appendix C.3). We observed that as the task becomes harder (from MNIST to CIFAR100), it also becomes harder to approximate the perfect $ucc$ classifier. Moreover, $UCC$ and $UCC_{\alpha=1}$ models, in general, have higher scores than their counterpart models of $UCC^{2+}$ and $UCC^{2+}_{\alpha=1}$, which is expected since the $ucc$ prediction task becomes easier at the absence of pure sets and models reach to early stopping condition (Appendix C.1) more easily. This is also supported by annother interesting, yet reasonable, observation that $UCC^{2+}$ models have higher $ucc$ accuracies than $UCC^{2+}_{\alpha=1}$ models thanks to the autoencoder branch which makes $UCC^{2+}$ harder to reach to early stopping condition.

## 4.3 Labels on Sets

Jensen-Shannon (JS) divergence (Lin, 1991) value between feature distributions of two pure sets consisting of instances of two different underlying classes is defined as inter-class JS divergence in this paper and used for comparison on 'Labels on sets' objective of assigning labels to pure sets. Higher values of inter-class JS divergence are desired since it means that feature distributions of

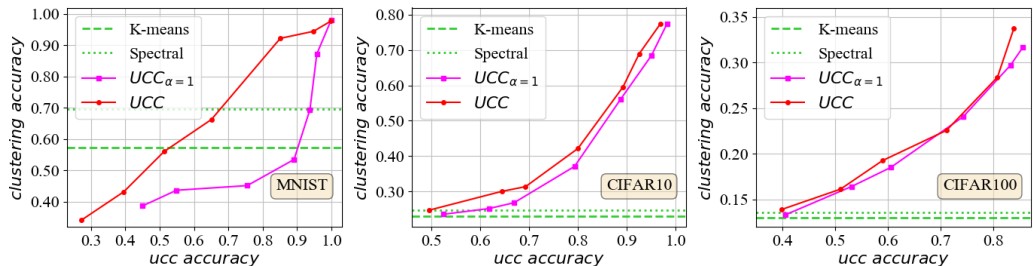

Figure 3: Clustering accuracy vs $ucc$ accuracy plots of $UCC$ and $UCC_{\alpha=1}$ models together with k-means and spectral clustering accuracy baselines on MNIST, CIFAR10 and CIFAR100 datasets.

pure sets of underlying classes are far apart from each other. The features of all the instances in a particular class are extracted by using a trained model and feature distributions associated to that class obtained by performing kernel density estimation on these extracted features. Then, for each pair of classes, inter-class JS divergence values are calculated (Appendix C.4). For a particular model, which is used in feature extraction, the minimum of these pairwise inter-class JS divergence values is used as a metric in the comparison of models. We have observed that as the task gets more challenging and the number of clusters increases, there is a drop in minimum inter-class JS divergence values, which is summarized in Table 1.

## 4.4 LABELS ON INSTANCES

For our eventual objective of 'Labels on instances', we have used 'clustering accuracy' as a comparison metric, which is calculated similar to Ghasedi Dizaji et al. (2017). By using our trained models, we extracted features of individual instances of all classes in test sets. Then, we performed unsupervised clustering over these features by using k-means and spectral clustering. We used number of classes in ground truth as number of clusters (MNIST: 10, CIFAR10: 10, CIFAR100: 20 clusters) during clustering and gave the best clustering accuracy for each model in Table 1 (Appendix C.5).

In Table 1, we compare clustering accuracies of our models together with baseline and state of the art models in the literature: baseline unsupervised (K-means (Wang et al., 2015), Spectral Clustering (Zelnik-Manor & Perona, 2005)); state of the art unsupervised (JULE (Yang et al., 2016), GMVAE (Dilokthanakul et al., 2016), DAC (Chang et al., 2017), DEPICT (Ghasedi Dizaji et al., 2017), DEC (Xie et al., 2016)) and state of the art semi-supervised (AAE (Makhzani et al., 2015), CatGAN (Springenberg, 2015), LN (Rasmus et al., 2015), ADGM (Maaløe et al., 2016)). Clustering performance of our *unique class count* models is better than the performance of unsupervised models in all datasets and comparable to performance of fully supervised learning models in MNIST and CIFAR10 datasets. The performance gap gets larger in CIFAR100 dataset as the task becomes harder. Although semi-supervised methods use some part of the dataset with 'exact' labels during training, our models perform on par with AAE and CatGAN models and comparable to LN and ADGM models on MNIST dataset. ADGM and LN even reach to the performance of the $FullySupervised$ model since they exploit training with 'exact' labeled data. On CIFAR10 dataset, LN and CatGAN models are slightly better than our *unique class count* models; however, they use 10% of instances with 'exact' labels, which is not a small portion.

In general, our $UCC$ and $UCC_{\alpha=1}$ models have similar performance, and they are better than their counterpart models of $UCC^{2+}$ and $UCC^{2+}_{\alpha=1}$ due to the absence of pure sets during training. However, in the real world tasks, the absence of pure sets heavily depends on the nature of the problem. In our task of semantic segmentation of breast cancer metastases in histological lymph node sections, for example, there are many pure sets. Furthermore, we observed that there is a performance gap between $UCC^{2+}$ and $UCC^{2+}_{\alpha=1}$ models: $UCC^{2+}$ models perform better than $UCC^{2+}_{\alpha=1}$ models thanks to the autoencoder branch. The effect of autoencoder branch is also apparent in Figure 3, which shows clustering accuracy vs $ucc$ accuracy curves for different datasets. For MNIST dataset, while $UCC$ model gives clustering accuracy values proportional to $ucc$ accuracy, $UCC_{\alpha=1}$ model cannot reach to high clustering accuracy values until it reaches to high $ucc$ accuracies. The reason is that autoencoder branch in $UCC$ helps $\theta_{\text{feature}}$ module to extract better features during the initial phases of the training process, where the $ucc$ classification accuracy is low. Compared to other

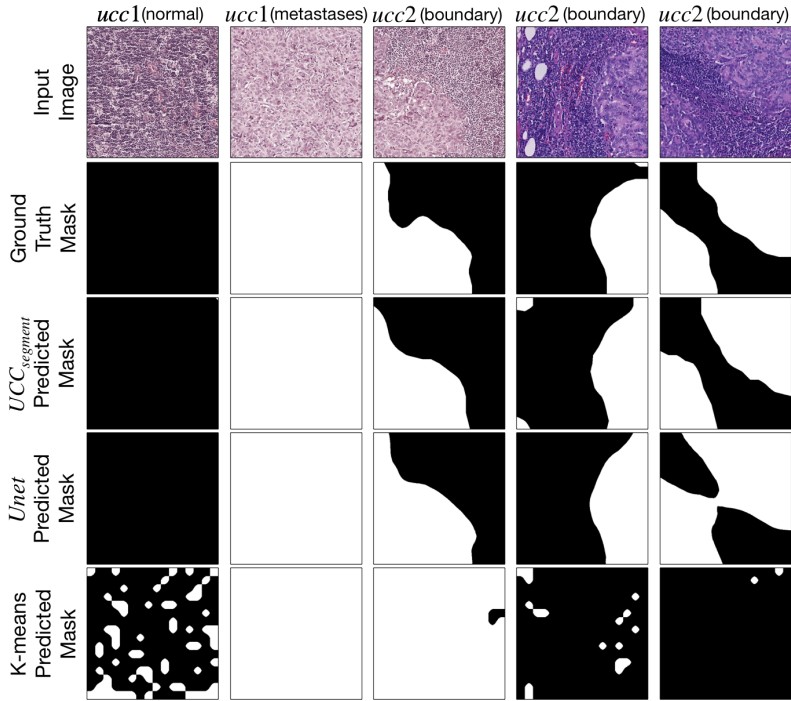

Figure 4: Example images from hold-out test dataset with corresponding $ucc$ labels, ground truth masks and predicted masks by $UCC_{segment}$, $Unet$ and K-means clustering models.

datasets, this effect is more significant in MNIST dataset since itself is clusterable. Although autoencoder branch helps in CIFAR10 and CIFAR100 datasets as well, improvements in clustering accuracy coming from autoencoder branch seems to be limited, so two models $UCC$ and $UCC_{\alpha=1}$ follow nearly the same trend in the plots. The reason is that CIFAR10 and CIFAR100 datasets are more complex than MNIST dataset, so autoencoder is not powerful enough to contribute to extract discrimant features, which is also confirmed by the limited improvements of $Autoencoder$ models over baseline performance in these datasets.

## 5 SEMANTIC SEGMENTATION OF BREAST CANCER METASTASES

Semantic segmentation of breast cancer metastases in histological lymph node sections is a crucial step in staging of breast cancer, which is the major determinant of the treatment and prognosis (Brierley et al., 2016). Given the images of lymph node sections, the task is to detect and locate, i.e. semantically segment out, metastases regions in the images. We have formulated this task in our novel MIL framework such that each image is treated as a bag and corresponding $ucc$ label is obtained based on whether the image is from fully normal or metastases region, which is labeled by $ucc1$, or from boundary region (i.e. image with both normal and metastases regions), which is labeled by $ucc2$. We have shown that this segmentation task can be achieved by using our weakly supervised clustering framework without knowing the ground truth metastases region masks of images, which require experts to exhaustively annotate each metastases region in each image. This annotation process is tedious, time consuming and more importantly not a part of clinical workflow.

We have used $512 \times 512$ image crops from publicly available CAMELYON dataset (Litjens et al., 2018) and constructed our bags by using $32 \times 32$ patches over these images. We trained our *unique class count* model $UCC_{segment}$ on $ucc$ labels. Then, we used the trained model as a feature extractor and conducted unsupervised clustering over the patches of the images in the hold-out test dataset to obtain semantic segmentation masks. For benchmarking purposes, we have also trained a fully supervised $Unet$ model (Ronneberger et al., 2015), which is a well-known biomedical image segmentation architecture, by using the ground truth masks and predicted the segmentation maps in

Table 2: Semantic segmentation performance statistics of $UCC_{segment}$, $Unet$ and K-means clustering methods on hold-out test dataset.

|  | TPR | FPR | TNR | FNR | PA |
|---|---|---|---|---|---|
| $UCC_{segment}$ (weakly supervised) | 0.818 | 0.149 | 0.851 | 0.182 | 0.863 |
| $Unet$ (fully supervised) | 0.860 | 0.126 | 0.874 | 0.140 | 0.889 |
| K-means (unsupervised baseline) | 0.370 | 0.271 | 0.729 | 0.630 | 0.512 |

the test set. The aim of this comparison was to show that at the absence of ground truth masks, our model can approximate the performance of a fully supervised model. Moreover, we have obtained semantic segmentation maps in the test dataset by using k-means clustering as a baseline study. Example images from test dataset with corresponding ground truth masks, $ucc$ labels and predicted masks by different models are shown in Figure 4. (Please see Appendix D.1 for more details.)

Furthermore, we have calculated pixel level gross statistics of TPR (True Positive Rate), FPR (False Positive Rate), TNR (True Negative Rate), FNR (False Negative Rate) and PA (Pixel Accuracy) over the images of hold-out test dataset and declared the mean values in Table 2 (Appendix D.2). When we look at the performance of unsupervised baseline method of K-means clustering, it is obvious that semantic segmentation of metastases regions in lymph node sections is not an easy task. Baseline method achieves a very low TPR value of 0.370 and almost random score of 0.512 in PA. On the other hand, both our weakly supervised model $UCC_{segment}$ and fully supervised model $Unet$ outperform the baseline method. When we compare our model $UCC_{segment}$ with $Unet$ model, we see that both models behave similarly. They have reasonably high TPR and TNR scores, and low FPR and FNR scores. Moreover, they have lower FPR values than FNR values, which is more favorable than vice-versa since pathologists opt to use immunohistochemistry (IHC) to confirm negative cases (Bejnordi et al., 2017). However, there is a performance gap between two models, which is mainly due to the fact that $Unet$ model is a fully supervised model and it is trained on ground truth masks, which requires exhaustive annotations by experts. On the contrary, $UCC_{segment}$ model is trained on $ucc$ labels and approximates to the performance of the $Unet$ model. $ucc$ label is obtained based on whether the image is metastatic, non-metastatic or mixture, which is much cheaper and easier to obtain compared to exhaustive mask annotations. Another factor affecting the performance of $UCC_{segment}$ model is that $ucc1$ labels can sometimes be noisy. It is possible to have some small portion of normal cells in cancer regions and vice-versa due to the nature of the cancer. However, our $UCC_{segment}$ is robust to this noise and gives reasonably good results, which approximates the performance of $Unet$ model.

## 6 CONCLUSION

In this paper, we proposed a weakly supervised learning based clustering framework and introduce a novel MIL task as the core of this framework. We defined $ucc$ as a bag level label in MIL setup and mathematically proved that a perfect $ucc$ classifier can be used to perfectly cluster individual instances inside the bags. We designed a neural network based $ucc$ classifer and experimentally showed that clustering performance of our framework with our $ucc$ classifiers are better than the performance of unsupervised models and comparable to performance of fully supervised learning models. Finally, we showed that our weakly supervised *unique class count* model, $UCC_{segment}$, can be used for semantic segmentation of breast cancer metastases in histological lymph node sections. We compared the performance of our model $UCC_{segment}$ with the performance of a $Unet$ model and showed that our weakly supervised model approximates the performance of fully supervised $Unet$ model. In the future, we want to check the performance of our $UCC_{segment}$ model with other medical image datasets and use it to discover new morphological patterns in cancer that had been overlooked in traditional pathology workflow.

## ACKNOWLEDGEMENTS

This work is supported by the Biomedical Research Council of the Agency for Science, Technology, and Research, Singapore and the National University of Singapore, Singapore.

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

# A    KERNEL DENSITY ESTIMATION

Kernel density estimation is a statistical method to estimate underlying unknown probability distribution in data (Parzen, 1962). It works based on fitting kernels at sample points of an unknown distribution and adding them up to construct the estimated probability distribution. Kernel density estimation process is illustrated in Figure 5.

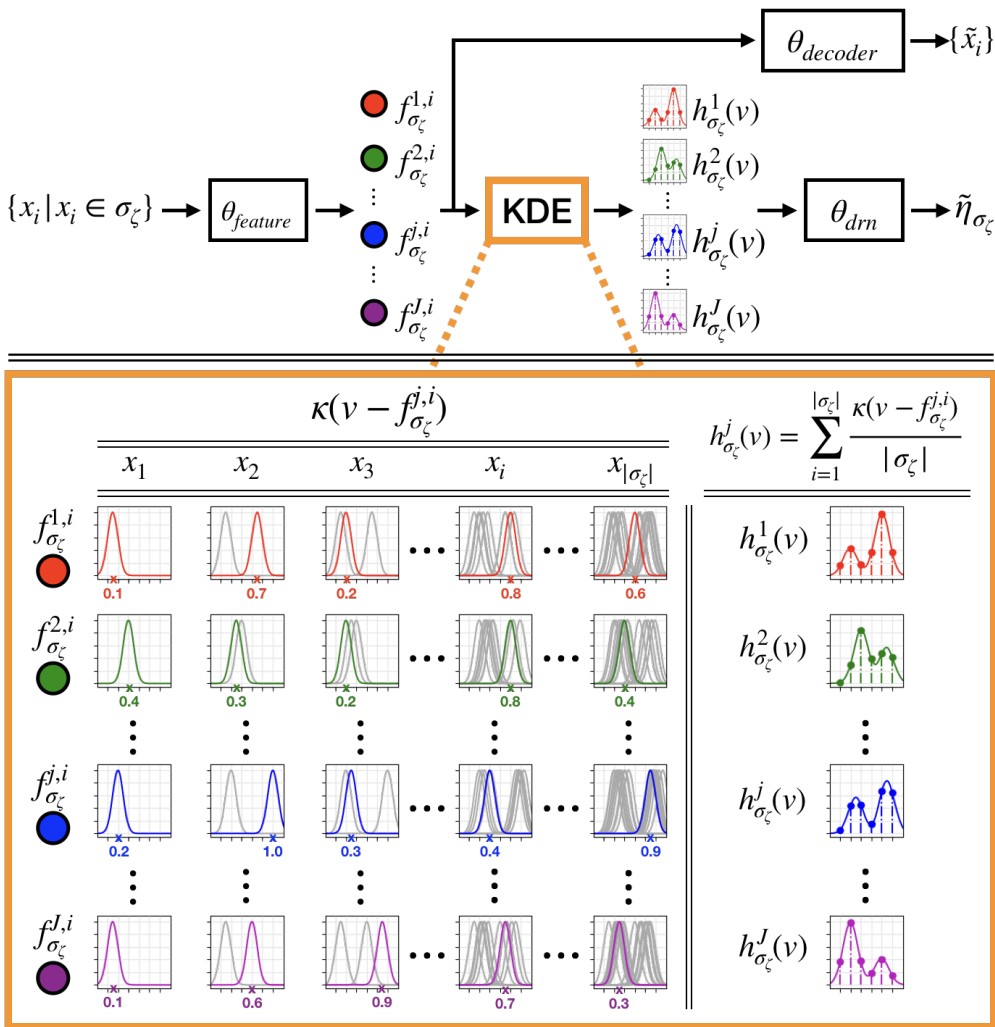

Figure 5: KDE module - the Gaussian kernel ($\kappa(v - f_{\sigma_\zeta}^{j,i})$) for each extracted feature for a sample is illustrated with colored curves and previously accumulated kernels are shown in gray. Estimated feature distributions, which are obtained by employing Equation 2, are sampled at some pre-determined intervals and passed to $\theta_{\text{drn}}$.

## A.1    KDE MODULE IS DIFFERENTIABLE

The distribution of the feature $h_{\sigma_\zeta}^j(v)$ is obtained by applying kernel density estimation on the extracted features $f_{\sigma_\zeta}^{j,i}$ as in Equation 2. In order to be able to train our *unique class count model* end-to-end, we need to show that KDE module is differentiable, so that we can pass the gradients from $\theta_{\text{drn}}$ to $\theta_{\text{feature}}$ during back-propagation. Derivative of $h_{\sigma_\zeta}^j(v)$ with respect to input of KDE

module, $f_{\sigma_\varsigma}^{j,i}$, can be obtained as in Equation 3.

$$h_{\sigma_\varsigma}^j(v) = \frac{1}{|\sigma_\varsigma|} \sum_{i=1}^{|\sigma_\varsigma|} \frac{1}{\sqrt{2\pi\sigma^2}} e^{-\frac{1}{2\sigma^2}\left(v - f_{\sigma_\varsigma}^{j,i}\right)^2} \tag{2}$$

$$\frac{\partial h_{\sigma_\varsigma}^j(v)}{\partial f_{\sigma_\varsigma}^{j,i}} = \frac{1}{|\sigma_\varsigma|} \frac{\left(v - f_{\sigma_\varsigma}^{j,i}\right)}{\sigma^2\sqrt{2\pi\sigma^2}} e^{-\frac{1}{2\sigma^2}\left(v - f_{\sigma_\varsigma}^{j,i}\right)^2} \tag{3}$$

After showing that KDE module is differentiable, we can show the weight update process for $\theta_{\text{feature}}$ module in our model. Feature extractor module $\theta_{\text{feature}}$ is shared by both autoencoder branch and $ucc$ branch in our model. During back-propagation phase of the end-to-end training process, the weight updates of $\theta_{\text{feature}}$ comprise the gradients coming from both branches (Equation 5). Gradients coming from autoencoder branch follow the traditional neural network back-propagation flow through the convolutional and fully connected layers. Different than that, gradients coming from $ucc$ branch (Equation 6) also back-propagate through the custom KDE layer according to Equation 3.

$$Loss = \alpha \underbrace{Loss_{ucc}}_{\substack{\text{ucc} \\ \text{loss}}} + (1-\alpha) \underbrace{Loss_{ae}}_{\substack{\text{autoencoder} \\ \text{loss}}} \text{ where } \alpha \in [0,1] \tag{4}$$

$$\underbrace{\frac{\partial Loss}{\partial \theta_{\text{feature}}}}_{\substack{\text{gradients} \\ \text{for} \\ \theta_{\text{feature}}}} = \alpha \underbrace{\frac{\partial Loss_{ucc}}{\partial \theta_{\text{feature}}}}_{\substack{\text{gradients} \\ \text{from} \\ \text{ucc branch}}} + (1-\alpha) \underbrace{\frac{\partial Loss_{ae}}{\partial \theta_{\text{feature}}}}_{\substack{\text{gradients} \\ \text{from} \\ \text{autoencoder branch}}} \tag{5}$$

$$\frac{\partial Loss_{ucc}}{\partial \theta_{\text{feature}}} = \frac{\partial Loss_{ucc}}{\partial h_{\sigma_\varsigma}} \times \underbrace{\frac{\partial h_{\sigma_\varsigma}}{\partial f_{\sigma_\varsigma}}}_{\substack{\text{back-propagation} \\ \text{through} \\ \text{KDE layer}}} \times \frac{\partial f_{\sigma_\varsigma}}{\partial \theta_{\text{feature}}} \tag{6}$$

# B    PROOFS OF PROPOSITIONS

Before proceeding to the formal proofs, it is helpful to emphasize the decomposability property of kernel density estimation here.

For any set, $\sigma_\varsigma$, one could partition it into a set of $M$ disjoint subsets $\sigma_\varsigma = \sigma_1' \cup \sigma_2' \cup \cdots \cup \sigma_M'$ where $\sigma_\lambda' \cap \sigma_\psi' = \emptyset$ for $\lambda \neq \psi$. It is trivial to show that distribution $h_{\sigma_\varsigma}^j(v)$ is simply a linear combination of distributions $h_{\sigma_\lambda'}^j(v), \lambda = 1, 2, \cdots, M$ (Equation 7). As a direct consequence, one could decompose any set into its pure subsets. This is an important decomposition which will be used in the proofs of propositions later.

$$h_{\sigma_\varsigma}^j(v) = \sum_{\lambda=1}^M w_{\sigma_\lambda'} h_{\sigma_\lambda'}^j(v), \forall j \text{ where } w_{\sigma_\lambda'} = \frac{|\sigma_\lambda'|}{|\sigma_\varsigma|} \tag{7}$$

Now, we can proceed to formally state our propositions.

**Definition 1** *Given a subset $\sigma_\varsigma \subset \mathcal{X}$, unique class count, $\eta_{\sigma_\varsigma}$, is defined as the number of unique classes that all instances in the subset $\sigma_\varsigma$ belong to, i.e. $\eta_{\sigma_\varsigma} = |\{\mathcal{L}(x_i)|x_i \in \sigma_\varsigma\}|$. Recall that each instance belongs to an underlying unknown class.*

**Definition 2** *A set $\sigma$ is called a pure set if its unique class count equals one. All pure sets are denoted by the symbol $\sigma^{pure}$ in this paper.*

**Proposition B. 1** *For any set $\sigma_\zeta \subset \mathcal{X}$, the unique class count $\eta_{\sigma_\zeta}$ of $\sigma_\zeta$ does not depend on the number of instances in $\sigma_\zeta$ belonging to a certain class.*

**Proof:** This conclusion is obvious from the definition of *unique class count* in Definition 1. ■

**Proposition B. 2** *$\theta_{drn}$ is non-linear.*

**Proof:** We give a proof by contradiction using Proposition B.1. Suppose $\theta_{drn}$ is linear, then

$$
\begin{aligned}
\theta_{drn}(h_{\sigma_\nu}) &= \theta_{drn}(w_\zeta h_{\sigma_\zeta} + w_\xi h_{\sigma_\xi}) \\
&= w_\zeta \theta_{drn}(h_{\sigma_\zeta}) + w_\xi \theta_{drn}(h_{\sigma_\xi}) \\
&= w_\zeta \tilde{\eta}_{\sigma_\zeta} + w_\xi \tilde{\eta}_{\sigma_\xi} = \tilde{\eta}_{\sigma_\nu}
\end{aligned}
\tag{8}
$$

Hence, $\theta_{drn}$ is linear only when Equation 8 holds. However, by Proposition B.1, $(\theta_{feature}, \theta_{drn})$ should count correctly regardless of the proportion of the size of the sets $|\sigma_\zeta|$ and $|\sigma_\xi|$. Hence, Equation 8 cannot hold true and $\theta_{drn}$ by contradiction cannot be linear. ■

**Proposition B. 3** *Let $\sigma_\zeta$, $\sigma_\xi$ be disjoint subsets of $\mathcal{X}$ with predicted unique class counts $\tilde{\eta}_{\sigma_\zeta}$ and $\tilde{\eta}_{\sigma_\xi}$, respectively. Let $\tilde{\eta}_{\sigma_\nu}$ be the predicted unique class count of $\sigma_\nu = \sigma_\zeta \cup \sigma_\xi$. If $h_{\sigma_\zeta} = h_{\sigma_\xi}$, then $\tilde{\eta}_{\sigma_\nu} = \tilde{\eta}_{\sigma_\zeta} = \tilde{\eta}_{\sigma_\xi}$.*

**Proof:** The distribution of set $\sigma_\nu$ can be decomposed into distribution of subsets,

$$
\begin{aligned}
h_{\sigma_\nu} &= w_\zeta h_{\sigma_\zeta} + w_\xi h_{\sigma_\xi} \text{ where } w_\zeta + w_\xi = 1 \tag{9} \\
h_{\sigma_\zeta} &= h_{\sigma_\xi} \implies h_{\sigma_\nu} = h_{\sigma_\zeta} \tag{10}
\end{aligned}
$$

Hence, $\tilde{\eta}_{\sigma_\nu} = \tilde{\eta}_{\sigma_\zeta} = \tilde{\eta}_{\sigma_\xi}$. ■

**Proposition 1** *Let $\sigma_\zeta$, $\sigma_\xi$ be disjoint subsets of $\mathcal{X}$ with predicted unique class counts $\tilde{\eta}_{\sigma_\zeta} = \tilde{\eta}_{\sigma_\xi} = 1$. If the predicted unique class count of $\sigma_\nu = \sigma_\zeta \cup \sigma_\xi$ is $\tilde{\eta}_{\sigma_\nu} = 2$, then $h_{\sigma_\zeta} \neq h_{\sigma_\xi}$.*

**Proof:** Proof of this proposition follows immediately from the contra-positive of Proposition B.3. ■

**Definition 3** *A perfect unique class count classifier takes in any set $\sigma$ and output the correct predicted unique class count $\tilde{\eta}_\sigma = \eta_\sigma$.*

**Proposition 2** *Given a perfect unique class count classifier. The dataset $\mathcal{X}$ can be perfectly clustered into $K$ subsets $\sigma_\xi^{pure}, \xi = 1, 2, \cdots, K$, such that $\mathcal{X} = \bigcup_{\xi=1}^{K} \sigma_\xi^{pure}$ and $\sigma_\xi^{pure} = \{x_i | x_i \in \mathcal{X}, \mathcal{L}(x_i) = \xi\}$.*

**Proof:** First note that this proposition holds because the "perfect unique class count classifier" is a very strong condition. Decompose $\mathcal{X}$ into subsets with single instance and then apply the unique class count on each subset, by definition, unique class counts of all subsets are one. Randomly pair up the subsets and merge them if their union still yield unique class count of one. Recursively apply merging on this condition until no subsets can be merged. ■

**Proposition 3** *Given a perfect unique class count classifier. Decompose the dataset $\mathcal{X}$ into $K$ subsets $\sigma_\xi^{pure}, \xi = 1, \cdots K$, such that $\sigma_\xi^{pure} = \{x_i | x_i \in \mathcal{X}, \mathcal{L}(x_i) = \xi\}$. Then, $h_{\sigma_\xi^{pure}} \neq h_{\sigma_\zeta^{pure}}$ for $\xi \neq \zeta$.*

**Proof:** Since in Proposition 1, the subsets are arbitrary, it holds for any two subsets with unique class count of one. By pairing up all combinations, one arrives at this proposition. Note that for a perfect unique class count classifier, $\eta = \tilde{\eta}$. ■

## C  DETAILS ON EXPERIMENTS WITH MNIST AND CIFAR DATASETS

### C.1  DETAILS OF MODEL ARCHITECTURES

Feature extractor module $\theta_{feature}$ has convolutional blocks similar to the wide residual blocks in Zagoruyko & Komodakis (2016). However, the parameters of architectures, number of convolutional and fully connected layers, number of filters in convolutional layers, number of nodes in

fully-connected layers, number of bins and $\sigma$ value in KDE module, were decided based on models' performance and training times. While increasing number of convolutional layers or filters were not improving performance of the models substantialy, they were putting a heavy computation burden. For determining the architecture of $\theta_{\text{drn}}$, we checked the performances of different number of fully connected layers. As the number of layers increased, the $ucc$ classification performance of the models increased. However, we want $\theta_{\text{feature}}$ to be powerful, so we stopped to increase number of layers as soon as we got good results. For KDE module, we have tried parameters of 11 bins, 21 bins, $\sigma = 0.1$ and $\sigma = 0.01$. Best results were obtained with 11 bins and $\sigma = 0.1$. Similarly, we have tested different number of features at the output of $\theta_{\text{feature}}$ module and we decided to use 10 features for MNIST and CIFAR10 datasets and 16 features for CIFAR100 dataset based on the clustering performance and computation burden.

During training, loss value of validation sets was observed as early stopping criteria. Training of the models was stopped if the validation loss didn't drop for some certain amount of training iterations.

For the final set of hyperparameters and details of architectures, please see the code for our experiments: `http://bit.ly/uniqueclasscount`

## C.2 DETAILS OF DATASETS

We trained and tested our models on MNIST, CIFAR10 and CIFAR100 datasets. While MNIST and CIFAR10 datasets have 10 classes, CIFAR100 dataset has 20 classes. For MNIST, we randomly splitted 10,000 images from training set as validation set, so we had 50,000, 10,000 and 10,000 images in our training $\mathcal{X}_{mnist,tr}$, validation $\mathcal{X}_{mnist,val}$ and test sets $\mathcal{X}_{mnist,test}$, respectively. In CIFAR10 dataset, there are 50,000 and 10,000 images with equal number of instances from each class in training and testing sets, respectively. Similar to MNIST dataset, we randomly splitted 10,000 images from the training set as validation set. Hence, we had 40,000, 10,000 and 10,000 images in our training $\mathcal{X}_{cifar10,tr}$, validation $\mathcal{X}_{cifar10,val}$ and testing $\mathcal{X}_{cifar10,test}$ sets for CIFAR10, respectively. In CIFAR100 dataset, there are 50,000 and 10,000 images with equal number of instances from each class in training and testing sets, respectively. Similar to other datasets, we randomly splitted 10,000 images from the training set as validation set. Hence, we had 40,000, 10,000 and 10,000 images in our training $\mathcal{X}_{cifar100,tr}$, validation $\mathcal{X}_{cifar100,val}$ and testing $\mathcal{X}_{cifar100,test}$ sets for CIFAR10, respectively.

$FullySupervised$ models took individual instances as inputs and were trained on instance level ground truths. $\mathcal{X}_{mnist,tr}$, $\mathcal{X}_{cifar10,tr}$ and $\mathcal{X}_{cifar100,tr}$ were used for training of $FullySupervised$ models. *Unique class count* models took sets of instances as inputs and were trained on $ucc$ labels. Inputs to *unique class count* models were sampled from the power sets of MNIST, CIFAR10 and CIFAR100 datasets, i.e. $2^{\mathcal{X}_{mnist,tr}}$, $2^{\mathcal{X}_{cifar10,tr}}$ and $2^{\mathcal{X}_{cifar100,tr}}$. For MNIST and CIFAR10 datasets, the subsets (bags) with 32 instances and for CIFAR100 dataset, the subsets (bags) with 128 instances are used in our experiments. While $UCC$ and $UCC_{\alpha=1}$ models are trained on $ucc1$ to $ucc4$ labels, $UCC^{2+}$ and $UCC^{2+}_{\alpha=1}$ models are trained on $ucc2$ to $ucc4$ labels.

Our models were trained on $ucc$ labels up to $ucc4$ instead of $ucc10$ ($ucc20$ in CIFAR100) since the performance was almost the same for both cases in our experiment with MNIST dataset, results of which are shown in Table 3. On the other hand, training with $ucc1$ to $ucc4$ was much faster than $ucc1$ to $ucc10$ because as the $ucc$ label gets larger, the number of instances in a bag is required to be larger in order to represent each class and number of elements in powerset also grows exponentially. Please note that for perfect clustering of instances, it is enough to have a perfect $ucc$ classifier that can discriminate $ucc1$ and $ucc2$ from Proposition 2.

All the results presented in this paper were obtained on hold-out test sets $\mathcal{X}_{mnist,test}$, $\mathcal{X}_{cifar10,test}$ and $\mathcal{X}_{cifar100,test}$.

Table 3: Clustering accuracy comparison of training *unique class count models* with $ucc$ labels of $ucc1$ to $ucc4$ and $ucc1$ to $ucc10$ on MNIST dataset.

|  | clustering accuracy | |
|---|---|---|
|  | $ucc1$ to $ucc4$ | $ucc1$ to $ucc10$ |
| $UCC$ | 0.984 | 0.983 |
| $UCC^{2+}$ | 0.984 | 0.982 |

### C.3 CONFUSION MATRICES FOR $ucc$ PREDICTIONS

We randomly sampled subsets for each $ucc$ label from the power sets of test sets and predicted the $ucc$ labels by using trained models. Then, we calculated the $ucc$ prediction accuracies by using predicted and truth $ucc$ labels, which are summarized in Table 1. Here, we show confusion matrices of our $UCC$ and $UCC^{2+}$ models on MNIST, CIFAR10 and CIFAR100 datasets as examples in Figure 6, 7 and 8, respectively.

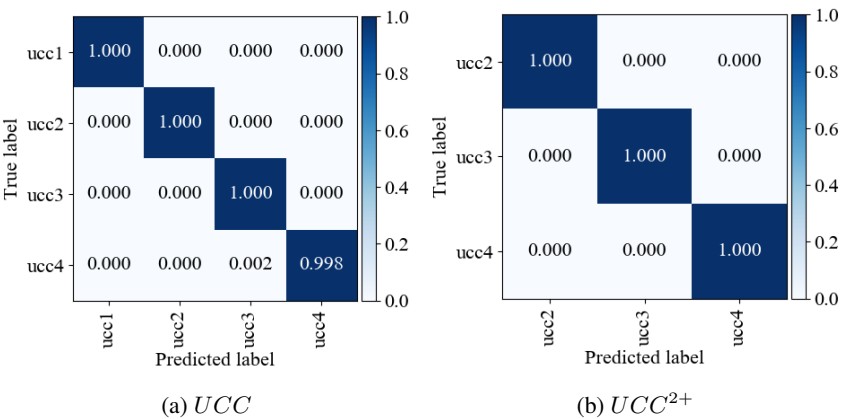

(a) $UCC$         (b) $UCC^{2+}$

Figure 6: Confusion matrices of our $UCC$ and $UCC^{2+}$ models for $ucc$ prediction on MNIST.

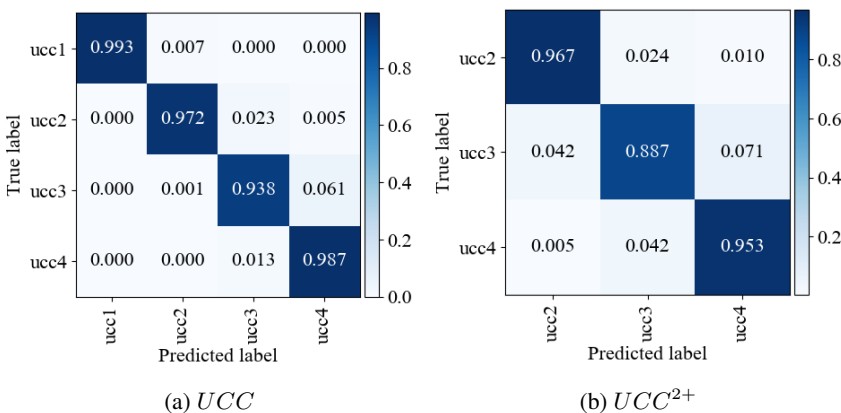

(a) $UCC$         (b) $UCC^{2+}$

Figure 7: Confusion matrices of our $UCC$ and $UCC^{2+}$ models for $ucc$ prediction on CIFAR10.

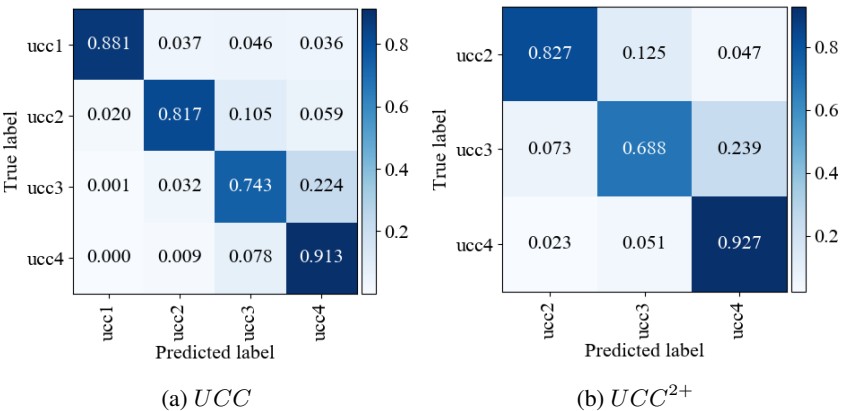

(a) $UCC$  (b) $UCC^{2+}$

Figure 8: Confusion matrices of our $UCC$ and $UCC^{2+}$ models for $ucc$ prediction on CIFAR100.

C.4    FEATURE DISTRIBUTIONS AND INTER-CLASS JS DIVERGENCE MATRICES

The features of all the instances in a particular class are extracted by using a trained model and feature distributions associated to that class obtained by performing kernel density estimation on these extracted features. Then, for each pair of classes, inter-class JS divergence values are calculated. We show inter-class JS divergence matrices for our $FullySupervised$ and $UCC$ models on MNIST test dataset in Figure 9. We also show the underlying distributions for $FullySupervised$ and $UCC$ models in Figure 10 and 11, respectively.

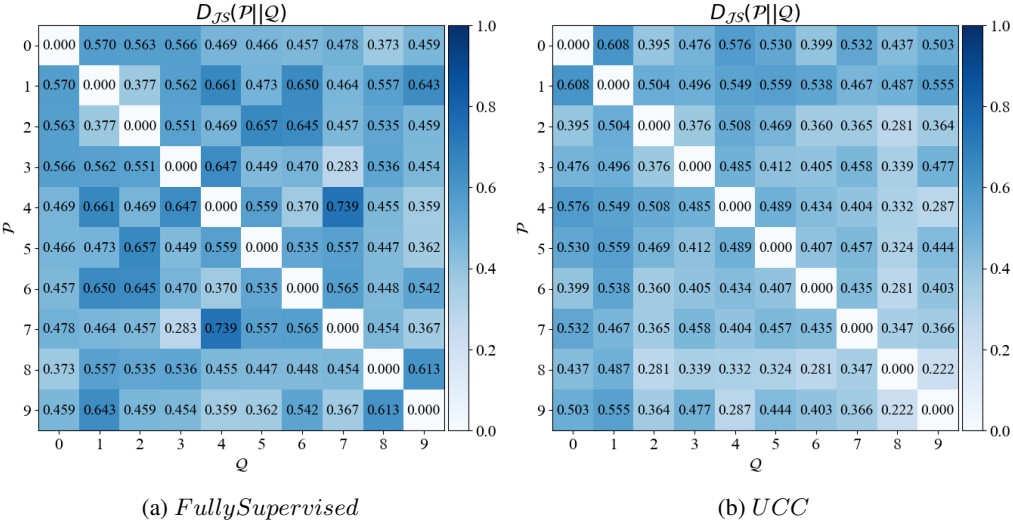

(a) $FullySupervised$  (b) $UCC$

Figure 9: Inter-class JS divergence matrix calculated over the distributions of features extracted by our $FullySupervised$ and $UCC$ models on MNIST test dataset.

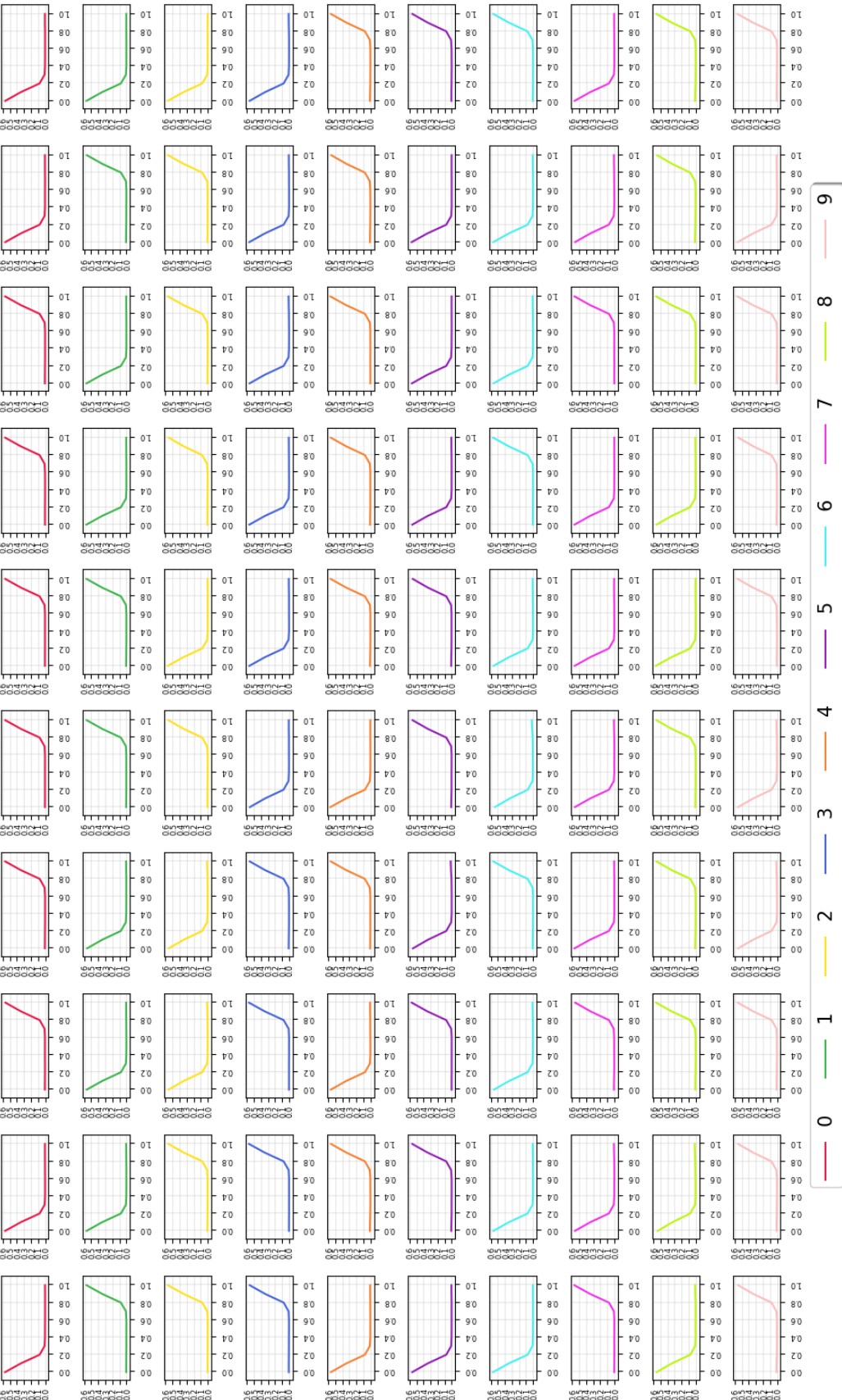

Figure 10: Distributions of extracted features by our $FullySupervised$ model on MNIST test dataset. Each column corresponds to a feature learned by model and each row corresponds to an underlying class in the test dataset.

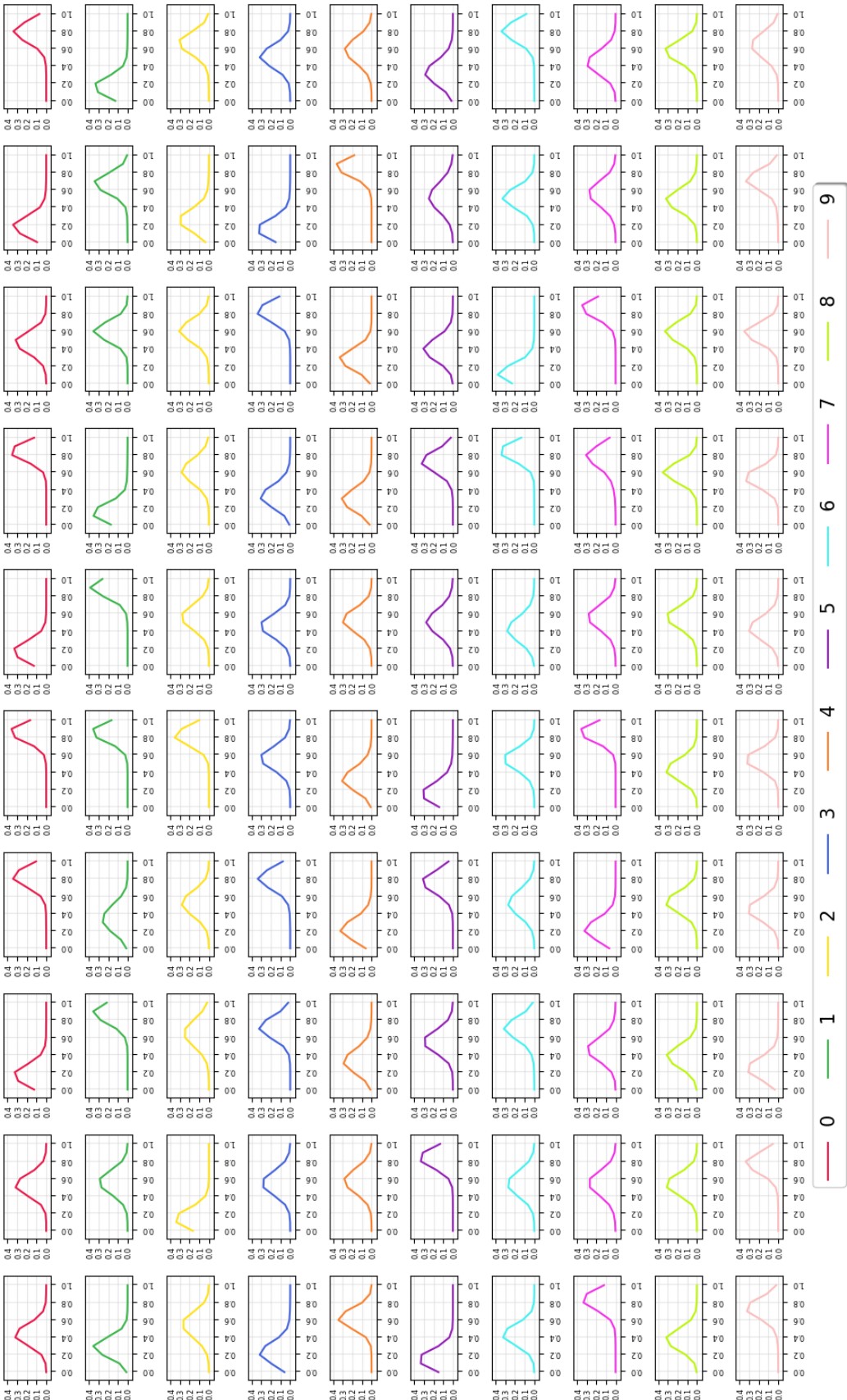

Figure 11: Distributions of extracted features by our $UCC$ model on MNIST test dataset. Each column corresponds to a feature learned by model and each row corresponds to an underlying class in the test dataset.

## C.5  K-MEANS AND SPECTRAL CLUSTERING ACCURACIES OF OUR MODELS

We performed unsupervised clustering by using k-means and spectral clustering and gave the best clustering accuracy for each model on each dataset in Table 1 in the main text. Here, we present all the clustering accuracies for our models in Table 4.

Table 4: Clustering accuracy values of our models with K-means and Spectral clustering methods on different test datasets. Best value for each model in each dataset is highlighted in **bold.**

|  | MNIST | | CIFAR10 | | CIFAR100 | |
|---|---|---|---|---|---|---|
|  | **K-means** | **Spectral** | **K-means** | **Spectral** | **K-means** | **Spectral** |
| $UCC$ | 0.979 | **0.984** | **0.781** | 0.680 | **0.338** | 0.261 |
| $UCC^{2+}$ | 0.977 | **0.984** | **0.545** | 0.502 | **0.278** | 0.225 |
| $UCC_{\alpha=1}$ | 0.981 | **0.984** | **0.774** | 0.635 | **0.317** | 0.249 |
| $UCC^{2+}_{\alpha=1}$ | **0.881** | 0.832 | **0.521** | 0.463 | **0.284** | 0.237 |
| $Autoencoder$ | **0.930** | 0.832 | **0.241** | 0.230 | **0.167** | 0.140 |
| $FullySupervised$ | **0.988** | 0.106 | **0.833** | 0.464 | **0.563** | 0.328 |

## C.6  UCC MODELS WITH AVERAGING LAYER AND KDE LAYER

KDE layer is chosen as MIL pooling layer in $UCC$ model because of its four main properties, first three of which are essential for the proper operation of proposed framework and validity of the propositions in the paper:

1. KDE layer is permutation-invariant, i.e. the output of KDE layer does not depend on the permutation of its inputs, which is important for the stability of $\theta_{\text{drn}}$ module.

2. KDE layer is differentiable, so $UCC$ model can be trained end-to-end.

3. KDE layer has decomposability property which enables our theoretical analysis (Appendix B).

4. KDE layer enables $\theta_{\text{drn}}$ to fully utilize the information in the shape of the distribution rather than looking at point estimates of distribution.

Averaging layer (Wang et al., 2018) as an MIL pooling layer, which also has the first three properties, can be an alternative to KDE layer in $UCC$ model. We have conducted additional experiments by replacing KDE layer with 'averaging layer' and compare the clustering accuracy values of the models with averaging layer and the models with KDE layer in Table 5.

Table 5: Clustering accuracy values of the models with averaging layer and the models with KDE layer.

|  | clustering acc. | |
|---|---|---|
|  | mnist | cifar10 |
| $UCC$ (KDE layer) | 0.984 | 0.781 |
| $UCC$ (Averaging layer) | 0.987 | 0.638 |
| $UCC_{\alpha=1}$ (KDE layer) | 0.981 | 0.774 |
| $UCC_{\alpha=1}$ (Averaging layer) | 0.943 | 0.508 |

# D  DETAILS ON SEMANTIC SEGMENTATION TASK

## D.1  DETAILS OF MODEL AND DATASET

Our model $UCC_{segment}$ has the same architecture with the $UCC$ model in CIFAR10 dataset, but this time we have used 16 features. We have also constructed the $Unet$ model with the same blocks used in $UCC_{segment}$ model in order to ensure a fair comparison. The details of the models can be seen in our code: `http://bit.ly/uniqueclasscount`

We have used $512 \times 512$ image crops from publicly available CAMELYON dataset (Litjens et al., 2018). CAMELYON dataset is a public Whole Slide Image (WSI) dataset of histological lymph node sections. It also provides the exhaustive annotations for metastases regions inside the slides which enables us to train fully supervised models for benchmarking of our weakly supervised *unique class count model*.

We randomly crop $512 \times 512$ images over the WSIs of CAMELYON dataset and associate a $ucc$ label to each image based on whether it is fully metastases/normal ($ucc1$) or mixture ($ucc2$). We assigned $ucc$ labels based on provided ground truths since they are readily available. However, please note that in case no annotations provided, obtaining $ucc$ labels is much cheaper and easier compared to tedious and time consuming exhaustive metastases region annotations. We assigned $ucc1$ label to an image if the metastases region in the corresponding ground truth mask is either less than 20% (i.e. normal) or more than 80% (i.e metastases). On the other hand, we assigned $ucc2$ label to an image if the metastases region in the corresponding ground truth mask is more than 30% and less than 70% (i.e. mixture). Actually, this labeling scheme imitates the noise that would have been introduced if $ucc$ labeling had been done directly by the user instead of using ground truth masks. Beyond that, $ucc1$ labels in this task can naturally be noisy since it is possible to have some small portion of normal cells in cancer regions and vice-versa due to the nature of the cancer. In this way, we have constructed our segmentation dataset consisting of training, validation and testing sets. The images in training and validation sets are cropped randomly over the WSIs in training set of CAMELYON dataset and the images in testing set are cropped randomly over the test set of CAMELYON dataset. Then, the bags in our MIL dataset to train $UCC_{segment}$ model are constructed by using $32 \times 32$ patches over these images. Each bag contains 32 instances, where each instance is a $32 \times 32$ patch. The details of our segmentation dataset are shown in Table 6.

We have provided the segmentation dataset under "./data/camelyon/" folder inside our code folder. If you want to use this dataset for benchmarking purposes please cite our paper (referenced later) together with the original CAMELYON dataset paper of Litjens et al. (2018).

Table 6: Details of our segmentation dataset: number of WSIs used to crop the images in each set, number of images in each set and corresponding label distributions in each set

|  | $ucc1$ | | | $ucc2$ | | |
|---|---|---|---|---|---|---|
|  | normal | metastases | total | mixture | # of images | # of WSIs |
| Training | 461 | 322 | 783 | 310 | 1093 | 159 |
| Validation | 278 | 245 | 523 | 211 | 734 | 106 |
| Testing | 282 | 668 | 950 | 228 | 1178 | 126 |

We have given confusion matrix for $ucc$ predictions of our $UCC_{segment}$ model in Figure 12. For $Unet$ model, we have shown loss curves of traininig and validation sets during training in Figure 13.

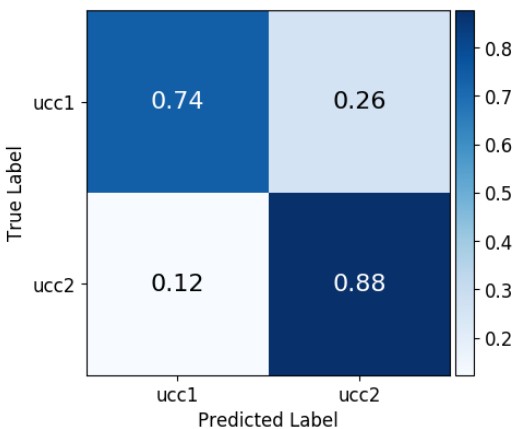

Figure 12: Confusion matrix of our $UCC_{segment}$ model for $ucc$ predictions on our segmentation dataset.

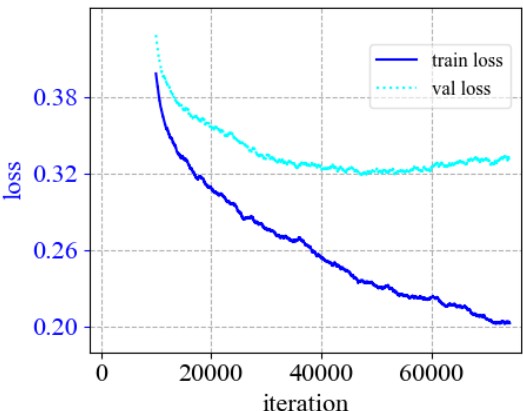

Figure 13: Training and validation loss curves during training of our $Unet$ model. We have used the best model weights, which were saved at iteration 58000, during training. Models starts to overfit after iteration 60000 and early stopping terminates the training.

### D.2 DEFINITIONS OF EVEALUTION METRICS

In this section, we have defined our pixel level evaluation metrics used for performance comparison of our weakly supervised $UCC_{segment}$ model, fully supervised $Unet$ model and unsupervised baseline $K-means$ model. Table 7 shows the structure of pixel level confusion matrix together with basic statistical terms. Then, our pixel level evaluation metrics TPR (True Positive Rate), FPR (False Positive Rate), TNR (True Negative Rate), FNR (False Negative Rate) and PA (Pixel Accuracy) are defined in Equation 11, 12, 13, 14 and 15, respectively.

Table 7: Structure of pixel level confusion matrix together with basic statistical terms

|  |  | Ground Truth | |
| --- | --- | --- | --- |
|  |  | **Positive (P)** | **Negative (N)** |
| **Predicted** | **Positive (P)** | True Positive (TP) | False Positive (FP) |
|  | **Negative (N)** | False Negative (FN) | True Negative (TN) |

$$TPR = \frac{TP}{TP + FN} \qquad (11)$$

$$FPR = \frac{FP}{FP + TN} \tag{12}$$

$$TNR = \frac{TN}{TN + FP} \tag{13}$$

$$FNR = \frac{FN}{FN + TP} \tag{14}$$

$$PA = \frac{TP + TN}{TP + FP + TN + FN} \tag{15}$$

