# OpenReview forum: "Weakly Supervised Clustering by Exploiting Unique Class Count"
_ICLR.cc/2020/Conference — Accept (Poster)_

### Official Review · AnonReviewer3 · 2019-10-22
**Official Blind Review #3**

**Rating:** 6

**Review:**

The authors present a novel weakly-supervised multiple instance learning model based on a bag level label called unique class count(UCC). The core content of the method is to learn mapping between bags and their associated bag level ucc labels and then to predict the ucc labels of unseen bags.
Positive:
(1) The authors use the unique class count(UCC) in the bag as a weak, bag level label to design a deep learning based UCC model for extracting features and clustering the individual instances in the unseen bags. They also construct a new optimization objective function by combining ucc loss and autoencoder loss. This is a novel weakly-supervised clustering method rarely seen.
(2) In this paper, a large number of experiments show the effectiveness of the algorithm in different data sets and semantic segmentation of breast cancer metastases.
Negative:
(1) The authors use KDE method to estimate the distribution of extracted features as the input of distribution regression module, but it is not clear how the description of KDE method updates on the parameters of the auto-encoder.
(2) The proofs of some propositions in Appendix B are very obvious and redundant. The authors’ proofs are not enough to prove whether the designed UCC classifier is perfect.

**Experience Assessment:**

I have published one or two papers in this area.

**Review Assessment: Checking Correctness Of Derivations And Theory:**

I assessed the sensibility of the derivations and theory.

**Review Assessment: Checking Correctness Of Experiments:**

I carefully checked the experiments.

**Review Assessment: Thoroughness In Paper Reading:**

I read the paper at least twice and used my best judgement in assessing the paper.

---

> ### Author Response · Authors · 2019-11-07
> **Request to clarify a few points in the review**
>
> Thank you very much for your valuable time and detailed review.
>
> In order to address the comments/concerns in the review in a better way, we want to clarify a few points in the review.
>
> Could you please kindly elaborate on the following statement from your review to help us to understand it better? Thanks.
>        "it is not clear how the description of KDE method updates on the parameters of the autoencoder."

---

> ### Author Response · Authors · 2019-11-09
> **Authors' Response to Reviewer 3, part 1**
>
> Thank you very much for your valuable time, insightful comments and suggestions. Please find our answers to your questions below. We have updated our paper based on suggestions and comments of the reviewers. The summary of updates in the paper are listed in a separate comment on top of the page. If you have any further comments or suggestions on the updated version of our paper, we will be glad to improve on them. Thanks.
>
> 1) "The authors use KDE method to estimate the distribution of extracted features as the input of distribution regression module, but it is not clear how the description of KDE method updates on the parameters of the autoencoder."
>
> --> Awaiting reviewer's reply to our request!
>       |--> Please see our response below.
>
> 2) "The proofs of some propositions in Appendix B are very obvious and redundant. The authors’ proofs are not enough to prove whether the designed UCC classifier is perfect."
>
> --> The reviewer is correct and we totally agree with the reviewer. Our proofs do not guarantee to find a perfect classifier (if a perfect classifier exists), nor do they prove UCC model is a universal approximator [1]. We will be glad to explore any idea and suggestion on this. However, from our propositions, we know that a perfect ucc classifier, theoretically, guarantees the perfect clustering of individual instances and in Figure 3, we experimentally showed that as ucc accuracy increases (i.e. as the classifier gets closer to the perfect), the clustering accuracy increases. Indeed, UCC model seems to be able to approximate the perfect ucc classifier in practice.
>
> [1] Kurt Hornik. Approximation capabilities of multilayer feedforward networks. Neural networks, 4 (2):251–257, 1991.

---

> > ### Author Response · Authors · 2019-11-12
> > **Authors' Response to Reviewer 3, part 2**
> >
> >
> > 1) "The authors use KDE method to estimate the distribution of extracted features as the input of distribution regression module, but it is not clear how the description of KDE method updates on the parameters of the autoencoder."
> >
> > --> Assuming that the concern of the reviewer was related to the weight updates of the ${\theta_{feature}}$ module, which is encoder part of autoencoder branch. We will explain the weight update procedure in here. If the reviewer has any other points to be clarified, we will be happy to address those points as well.
> >
> > Considering the reviewer's concern, we have updated Appendix A.1 to include the weight update procedure for the feature extractor module ${\theta_{feature}}$, which is summarised here as well.
> >
> > Feature extractor module ${\theta_{feature}}$ is shared by both autoencoder branch and $ucc$ branch in our model. During back-propagation phase of the end-to-end training process, the weight updates of ${\theta_{feature}}$ comprise the gradients coming from both branches (Equation (2)). Gradients coming from autoencoder branch follow the traditional neural network back-propagation flow through the convolutional and fully connected layers. Different than that, gradients coming from $ucc$ branch (Equation (3)) also back-propagate through the custom KDE layer according to the differential equation in Equation (4), derivation of which has been given in Appendix A.1.
> >
> > (1) $Loss\ =\ \alpha \underbrace{Loss_{ucc}}_{ \substack{\text{ucc} \\ \text{loss}} }\ +\ (1-\alpha)\underbrace{Loss_{ae}}_{ \substack{\text{autoencoder} \\ \text{loss}} } \text{ where } \alpha \in [0,1]$
> >
> > (2) $\underbrace{\frac{\partial Loss}{\partial {\theta_{feature}}}}_{ \substack{\text{gradients} \\ \text{for} \\ \theta_{feature}} }\ =\ \alpha \underbrace{\frac{\partial Loss_{ucc}}{\partial {\theta_{feature}}}}_{ \substack{\text{gradients} \\ \text{from} \\ \text{ucc branch}} }\ +\ (1-\alpha) \underbrace{\frac{\partial Loss_{ae}}{\partial {\theta_{feature}}}}_{\substack{\text{gradients} \\ \text{from} \\ \text{autoencoder branch}}}$
> >
> > (3) $\frac{\partial Loss_{ucc}}{\partial {\theta_{feature}}}=\frac{\partial Loss_{ucc}}{\partial h_{\sigma_\zeta}} \times \underbrace{\frac{\partial h_{\sigma_\zeta}}{\partial f_{\sigma_\zeta}}}_{ \substack{\text{back-propagation} \\ \text{through} \\ \text{KDE layer}} } \times \frac{\partial f_{\sigma_\zeta}}{\partial {\theta_{feature}}}$
> >
> > (4) $\frac{\partial h^j_{\sigma_\zeta}(v)}{\partial f^{j,i}_{\sigma_\zeta}} = \frac{1}{ |\sigma_\zeta| } \frac{\left(v-f^{j,i}_{\sigma_\zeta}\right)}{ {{\sigma}^2} {\sqrt{2\pi{\sigma}^2}} } e^{-\frac{1}{2{\sigma}^2}\left(v-f^{j,i}_{\sigma_\zeta}\right)^2}$
> >
> > $\sigma_\zeta$: a set of instances
> >
> > $h^j_{\sigma_\zeta}(v)$: distribution of $j^{th}$ feature obtained over the instances of set $\sigma_\zeta$ at the output of KDE layer
> >
> > $f^{j,i}_{\sigma_\zeta}$: $j^{th}$ feature of $i^{th}$ instance in set $\sigma_\zeta$ extracted by ${\theta_{feature}}$
> >
> > $\sigma$: standard deviation of Gaussian kernel in KDE module

---

### Official Review · AnonReviewer1 · 2019-10-22
**Official Blind Review #1**

**Rating:** 1

**Review:**

This paper proposes a MIL clustering method. The proposed MIL setup is called "unique class count (ucc)", this is, for a bag os samples ucc is the number of clusters in the bag. The method learns the features of the samples using two losses: an autoender loss and the ucc loss. Once trained on a dataset the method can perform clustering on classes (classify) better than a fully unsupervised clustering algorithm and worse than a fully supervised model. The method is evaluated on MNIST, CIFAR10, CIFAR100  and on binary breast cancer segmentation.

The main table of the paper (Table 1) compares the results of the proposed methods only with unsupervised methods and a fully supervised one. The results are reasonable. However, not a fair comparison. More fair comparisons can be found in Table 6 of appendix C.6 page 22 where the method is compared to semi-supervised methods. In this case, the proposed method performs worse than the semisupervised methods.

UCC model uses bags of sizes from 1 to 4. Assuming uniform distribution, 25% of the samples are fully labeled. The semi-supervised methods from Table 6 how many labeled samples do they use? Having that small bag samples the problem is quite easy. Moreover, during training, I guess that the same sample can go to different bags. Is this right? If so, the annotation time would be very big. And also, one could trivially get the full label of each sample.

The article is excessively long. 10 pages for the main article and 12 extra pages for supplementary material which are needed for understanding the paper. Many of the definitions are redundant and there is excessive detail in explanations that can be expressed more simply.



**Experience Assessment:**

I do not know much about this area.

**Review Assessment: Checking Correctness Of Derivations And Theory:**

I assessed the sensibility of the derivations and theory.

**Review Assessment: Checking Correctness Of Experiments:**

I carefully checked the experiments.

**Review Assessment: Thoroughness In Paper Reading:**

I read the paper at least twice and used my best judgement in assessing the paper.

---

> ### Author Response · Authors · 2019-11-07
> **Request to clarify a few points in the review**
>
> Thank you very much for your valuable time and detailed review.
>
> In order to address the comments/concerns in the review in a better way, we want to clarify a few points in the review.
>
> Could you please kindly help us to understand the following statements from your review better? Thanks.
>
> 1) "UCC model uses bags of sizes from 1 to 4."
> Does that mean the number of instances inside a bag is either 1 or 2 or 3 or 4?
>
> 2) "Having that small bag samples the problem is quite easy."
> What does "small bag samples" refer to?
>
> 3) "Once trained on a dataset the method can perform clustering on classes (classify) better than a fully unsupervised clustering algorithm and worse than a fully supervised model."
> What does "perform clustering on classes (classify)" mean?

---

> ### Author Response · Authors · 2019-11-09
> **Authors' Response to Reviewer 1, part 1**
>
> Thank you very much for your valuable time, insightful comments and suggestions. Please find our answers to your questions below. We have updated our paper based on suggestions and comments of the reviewers. The summary of updates in the paper are listed in a separate comment on top of the page. If you have any further comments or suggestions on the updated version of our paper, we will be glad to improve on them. Thanks.
>
> 1) "The main table of the paper (Table 1) compares the results of the proposed methods only with unsupervised methods and a fully supervised one. The results are reasonable. However, not a fair comparison. More fair comparisons can be found in Table 6 of appendix C.6 page 22 where the method is compared to semi-supervised methods. In this case, the proposed method performs worse than the semisupervised methods."
>
> --> Considering the reviewer's concern, in Table 1, we have included semi-supervised models as well. Previously, only the unsupervised and fully supervised models were included in Table 1 since they set the lower bound and upper bound on the performance scale bar, respectively. The aim was to show that UCC models performed somewhere in between those lower and upper bounds. Actually, the performance of UCC models were explicitly compared with the performance of semi-supervised models in the main text (Sec 4.4 Labels on Intances, paragraph 2) and the reader was referred to Appendix C.6 for detailed comparison due to space limitations. Now, everything is included in the main text.
>
> 2) "UCC model uses bags of sizes from 1 to 4. Assuming uniform distribution, 25\% of the samples are fully labeled. The semi-supervised methods from Table 6 how many labeled samples do they use? Having that small bag samples the problem is quite easy. Moreover, during training, I guess that the same sample can go to different bags. Is this right? If so, the annotation time would be very big. And also, one could trivially get the full label of each sample."
>
> --> Awaiting reviewer's reply to our request!
>       |--> Please see our response below.
>
> 3) "The article is excessively long. 10 pages for the main article and 12 extra pages for supplementary material which are needed for understanding the paper. Many of the definitions are redundant and there is excessive detail in explanations that can be expressed more simply."
>
> --> Following the reviewer's comment, we have gone through our paper again and moved some of the propositions, which are obvious and used in the proof of other propositions, from Sec 3.2.2 to Appendix B. We keep working on our paper to improve it further during the rebuttal period.
>
> 4) "Once trained on a dataset the method can perform clustering on classes (classify) better than a fully unsupervised clustering algorithm and worse than a fully supervised model."
>
> --> Awaiting reviewer's reply to our request!
>       |--> Please see our response below.

---

> > ### Author Response · Authors · 2019-11-12
> > **Authors' Response to Reviewer 1, part 2**
> >
> >
> > 2) "UCC model uses bags of sizes from 1 to 4. Assuming uniform distribution, 25% of the samples are fully labeled. The semi-supervised methods from Table 6 how many labeled samples do they use? Having that small bag samples the problem is quite easy. Moreover, during training, I guess that the same sample can go to different bags. Is this right? If so, the annotation time would be very big. And also, one could trivially get the full label of each sample."
> >
> > Assuming that the main concern of the reviewer was related to number of instances inside the bags, i.e. bag sizes, we have addressed the reviewer's concerns in here. If the reviewer has any other points to be clarified, we will be happy to address those points as well.
> >
> > - "UCC model uses bags of sizes from 1 to 4. (...) Having that small bag samples the problem is quite easy."
> > --> We apologize for misunderstanding. We didn't specify the number of instances inside the bags. Considering the reviewer's concern, we have updated Appendix C.2 to include number of instances in each bag for each dataset. For MNIST and CIFAR10 datasets, bags with 32 instances and for CIFAR100 dataset, bags with 128 instances were used in all experiments. We thank the reviewer for addressing this point.
> >
> > - "Assuming uniform distribution, 25% of the samples are fully labeled."
> > --> During training, labels of individual instances inside the bags are unknown (Sec. 3 Objective, Sec. 3.1 Definition 1). Even if a bag is a $ucc1$ bag, only thing we know is that all the instances inside the bag belongs to the same class, but the class is unknown. The unknown class, for example, can be anything in $\{0,1,\cdots,9\}$ for MNIST. Hence, we don't have any 'fully labeled' instances, i.e. we don't know the labels of individual instances.
> >
> > - "The semi-supervised methods from Table 6 how many labeled samples do they use?"
> > --> The semi-supervised models in Table 1 (previously in Table 6) use 100 instances with labels for MNIST dataset. Please note that MNIST is such an easy dataset that 100 instances with labels are enough to boost the performance of semi-supervised models. On the other hand, they use 10% of instances with labels for CIFAR10 dataset, which is not a small portion.
> >
> > - "Moreover, during training, I guess that the same sample can go to different bags. Is this right?"
> > --> The reviewer is correct: one instance may appear in more than one bag since inputs to $ucc$ models were sampled from the power sets of MNIST, CIFAR10 and CIFAR100 datasets, i.e. $2^{{\mathcal{X}}_{mnist,tr}}$, $2^{{\mathcal{X}}_{cifar10,tr}}$ and $2^{{\mathcal{X}}_{cifar100,tr}}$, during training (Sec. 4.1 paragraph 2).
> >
> > - "If so, the annotation time would be very big. And also, one could trivially get the full label of each sample."
> > --> We like to emphasize that $UCC$ model is a typical example of Multiple Instance Learning (MIL) paradigm and it is different than the traditional supervised classification paradigm, where individual instances are labeled.
> >
> > It is important to state that in almost all MIL problems, bag level labels are readily available and task is to infer instance level labels. Trying to obtain bag level labels from instance level labels is either not possible (instance level labels are not known) or not feasible (i.e. directly obtaining bag level labels is much cheaper). For example, in our histopathology example, labelling an image with $ucc1$ or $ucc2$ label is much cheaper than annotating each individual pixel (instance) inside the image.
> >
> > We used MNIST and CIFAR datasets in our experiments and obtained bag level labels from instance labels while preparing MIL dataset. However, this was just to construct controlled datasets to analyze the characteristics of our models properly. Moreover, individual instance labels were never used during training of our models.
> >
> > 4) "Once trained on a dataset the method can perform clustering on classes (classify) better than a fully unsupervised clustering algorithm and worse than a fully supervised model."
> > --> We like to highlight that clustering was performed on the features extracted by feature extractor module $\bar{\theta}_{feature}$ of trained models (Sec. 3.3).

---

### Official Review · AnonReviewer2 · 2019-10-23
**Official Blind Review #2**

**Rating:** 8

**Review:**

This paper proposes a new type of weakly supervised clustering / multiple instance learning (MIL) problem in which bags of instances (data points) are labeled with a "unique class count (UCC)*, rather than any bag-level or instance-level labels.  For example, a histopathology slide (the bag), consisting of many individual pixels to be labeled (the instances) could be labeled at the bag level only with UCC = 1 (for only healthy or only metastatic) or UCC = 2 (for mixed / border case).  The paper then proposes an approach for clustering instances based on the following two-step approach: (1) a UCC model is trained to predict the UCC given an input bag, and (2) the features of this learned UCC model are used in an unsupervised clustering algorithm to the get the instance-level clusters / labels.  The paper also provides a theoretical argument for why this approach is feasible.

Overall this paper proposes a (1) novel and creative approach that is well validated both (2) theoretically and (3) empirically, and relevant to a real-world problem, and thus I believe should be accepted.  In slightly more detail:

- (1) MIL where we are given bag-level labels only is a well-studied problem that occurs in many real world settings, such as the histopathology one used in this paper.  However to this reader's knowledge, this is a new variant that is both creative and motivated by an actual real-world study, which is exciting and alone warrants presentation at the conference in my opinion.

- (2) The theoretical treatment is high-level, but still serves a clear purpose of establishing feasibility of the proposed method- this modest and appropriate purpose serves the paper well.

- (3) The empirical results are thorough---e.g. the use of the two loss components for the UCC model are appropriately ablated, there are a range of baseline approaches compared to, multiple evaluation points are provided (i.e. both UCC prediction and final clustering metrics), and a real world use case is presented--and the results are impressive.

Some comments to improve the paper:
- The connection of the proposed UCC approach to the motivating histopath example should be explicitly stated upfront to help the reader understand how this method could be used and how it makes sense!
- It seems that the KDE element of the UCC model was chosen in part to enable the theoretical analysis?  If so, this should be clearly stated to help the reader understand the design rationale.
- Either way, it seems that a different approach than the KDE layer could have been taken- this should be added to the ablation experiments

**Experience Assessment:**

I have published one or two papers in this area.

**Review Assessment: Checking Correctness Of Derivations And Theory:**

I assessed the sensibility of the derivations and theory.

**Review Assessment: Checking Correctness Of Experiments:**

I assessed the sensibility of the experiments.

**Review Assessment: Thoroughness In Paper Reading:**

I read the paper thoroughly.

---

> ### Author Response · Authors · 2019-11-09
> **Authors' Response to Reviewer 2, part 1**
>
> Thank you very much for your valuable time, insightful comments and suggestions. Please find our answers to your questions below. We have updated our paper based on suggestions and comments of the reviewers. The summary of updates in the paper are listed in a separate comment on top of the page. If you have any further comments or suggestions on the updated version of our paper, we will be glad to improve on them. Thanks.
>
> 1) "The connection of the proposed UCC approach to the motivating histopath example should be explicitly stated upfront to help the reader understand how this method could be used and how it makes sense!"
>
> --> Following the reviewer's suggestion, we have stated the connection of the proposed UCC approach to the motivating histopathology example by introducing our formulation of semantic segmentation task in proposed framework. We have updated the second paragraph in page 2 starting with "Our weakly supervised clustering framework is illustrated in Figure 1. It consists ...".
>
> 2) "It seems that the KDE element of the UCC model was chosen in part to enable the theoretical analysis? If so, this should be clearly stated to help the reader understand the design rationale."
>
> --> This is a great observation and the reviewer is correct. One of the main reasons in choosing KDE module was to enable the theoretical analysis. Kernel density estimation has decomposability property, which was used in the proofs of propositions. Although this was highlighted at the beginning of Appendix B, we unfortunately did not include this point in the main text. Thanks for raising this point, we have updated Sec 3.2.1 accordingly.
>
> 3) "It seems that a different approach than the KDE layer could have been taken- this should be added to the ablation experiments"
>
> --> That is true. Instead of KDE layer, for example, an "averaging layer" [1], which gives the average value of feature scores accross the instances inside the bag,  could have been used in our proposed architecture while still all of the propositions in the paper are valid. We chose KDE over "averaging layer" since we simply wanted to provide the distribution regression module with the full distribution information rather than point estimates. Our concern was that two different distributions may have the same point estimates, which would make the training process harder.
>
> We designed and conducted a complete set of experiments with KDE module to compare different models in our paper. However, as the reviewer suggested, it could be good to see the performance of other approaches and compare with the performance of KDE layer. Although a complete set of experiments with different MIL pooling layers, different models and different datasets is not feasible to be done by rebuttal deadline, we agree with the reviewer on exploring the alternatives as much as we can within the rebuttal period. Hence, we have started to conduct experiments with "averaging layer" in some of our models. We have presented the results of completed experiments in Appendix C.6 and we will update this part once the results are available.
>
> [1] Xinggang Wang, Yongluan Yan, Peng Tang, Xiang Bai, and Wenyu Liu. Revisiting multiple instance neural networks. Pattern Recognition, 74:15–24, 2018.

---

### Author Response · Authors · 2019-11-09
**Summary of updates in the paper**

The updates made in the paper during rebuttal period are summarised in here!

- We have stated the connection of the proposed UCC approach to the motivating histopathology example by introducing our formulation of semantic segmentation task in proposed framework. We have updated the second paragraph in page 2 starting with "Our weakly supervised clustering framework is illustrated in Figure 1. It consists ...".

- We have updated Sec 3.2.1 to highlight that KDE module also enables our theoretical analysis beyond providing us with some other unique advantages.

- We have moved some of the propositions, which are obvious and used in the proof of other propositions, from Sec 3.2.2 to Appendix B.

- We have included semi-supervised models in Table 1 and removed from Appendix C.6

- We have updated Appendix A.1 to include weight update procedure for feature extractor module ${\theta_{feature}}$.

- We have updated Appendix C.2 to include number of instances in each bag for each dataset.

- We have devoted Appendix C.6 to include the clustering accuracy results of UCC models with "averaging layer" as an alternative to KDE layer.

---

### Decision · Program_Chairs · 2019-12-19

**Decision:**

Accept (Poster)

**Comment:**

The paper proposes a weakly supervised learning algorithm, motivated by its application to histopathology. Similar to the multiple instance learning scenario, labels are provided for bags of instances. However instead of a single (binary) label per bag, the paper introduces the setting where the training algorithm is provided with the number of classes in the bag (but not which ones). Careful empirical experiments on semantic segmentation of histopathology data, as well as simulated labelling from MNIST and CIFAR demonstrate the usefulness of the method. The proposed approach is similar in spirit to works such as learning from label proportions and UU learning (both which solve classification tasks).
http://www.jmlr.org/papers/volume10/quadrianto09a/quadrianto09a.pdf
https://arxiv.org/abs/1808.10585

The reviews are widely spread, with a low confidence reviewer rating (1). However it seems that the high confidence reviewers are also providing higher scores and better comments. The authors addressed many of the reviewer comments, and seeked clarification for certain points, but the reviewers did not engage further during the discussion period.

This paper provides a novel weakly supervised learning setting, motivated by a real world semantic segmentation task, and provides an algorithm to learn from only the number of classes per bag, which is demonstrated to work on empirical experiments. It is a good addition to the ICLR program.